# PROCEEDINGS A

atomic and molecular physics, applied mathematics, computational chemistry

diffusion maps, molecular dynamics, committors, metastability

**Author for correspondence:**
Z. Trstanova
e-mail: zofia.trstanova@gmail.com

# Local and global perspectives on diffusion maps in the analysis of molecular systems

Z. Trstanova[1], B. Leimkuhler[1] and T. Lelièvre[2]

[1]School of Mathematics, University of Edinburgh, James Clerk Maxwell Building, Peter Guthrie Tait Road, Edinburgh EH9 3FD, UK
[2]CERMICS (ENPC), INRIA, Marne-la-Vallée 77455, France

ZT, 0000-0002-1152-8930

Diffusion maps approximate the generator of Langevin dynamics from simulation data. They afford a means of identifying the slowly evolving principal modes of high-dimensional molecular systems. When combined with a biasing mechanism, diffusion maps can accelerate the sampling of the stationary Boltzmann–Gibbs distribution. In this work, we contrast the local and global perspectives on diffusion maps, based on whether or not the data distribution has been fully explored. In the global setting, we use diffusion maps to identify metastable sets and to approximate the corresponding committor functions of transitions between them. We also discuss the use of diffusion maps *within* the metastable sets, formalizing the locality via the concept of the quasi-stationary distribution and justifying the convergence of diffusion maps within a local equilibrium. This perspective allows us to propose an enhanced sampling algorithm. We demonstrate the practical relevance of these approaches both for simple models and for molecular dynamics problems (alanine dipeptide and deca-alanine).

## 1. Introduction

The calculation of thermodynamic averages for complex models is a fundamental challenge in computational chemistry [1], materials modelling [2] and biology [3]. In typical situations, the potential energy $U$ of the system is known, and the states are assumed

(*a*) (*b*) (*c*)

*A* *B*1 *B*2

**Figure 1.** (*a–c*) Folding of deca-alanine: three metastable conformations. (Online version in colour.)

to be distributed according to the Boltzmann–Gibbs distribution with density

$$\rho_\beta = Z^{-1}e^{-\beta U}.$$

The difficulty arises due to the high-dimensional, multimodal nature of the target distribution in combination with limited computational resources. The multimodality of the target distribution causes that the high-likelihood conformational states are separated by low-probability transition regions. In such a setting, the transitions between critical states become 'rare events', meaning that naive computational approaches converge slowly (if at all) or produce inaccurate results. Moreover, standard MD simulations of large proteins (more than 500 residues) run on regular CPUs allow to simulate dynamics over hundreds of nanoseconds (or a few micrometre max) in a reasonable amount of time. This time scale is often not enough to sample relevant conformational transitions which may occur on the millisecond time scale.

As an illustration, consider the problem of exploring all folded configurations of a deca-alanine molecule in vacuum at a specific temperature (300 K). This short peptide is a common model for illustrating the difficulties of folding due its very complicated free energy landscape [4]. Three typical states are shown in figure 1. The many interactions between atoms of the molecule mean that changes in structure are the result of a sequence of coordinated moves. Because the transitions between the states occur infrequently (they are 'rare events') the results of short Langevin dynamics simulation will typically constitute a highly localized exploration of the region surrounding one particular conformational minimum.

The goal of enhanced sampling strategies is to dramatically expand the range of observed states by modifying in some way the dynamical model. Typical schemes [5–11] rely on the definition of collective variables (CVs) in order to drive the enhanced sampling of the system only in relevant low-dimensional coordinates describing the slowest time scales.

In this article, we discuss the automatic identification of CVs for the purpose of enhancing sampling in applications such as molecular dynamics. In some cases, the natural CVs relate to underlying physical processes and can be chosen using scientific intuition, but in many cases, this is far from straightforward, as there may be competing molecular mechanisms underpinning a given conformational change. Methods capable of automatically detecting CVs have, moreover, much wider potential for application. Many Bayesian statistical inference calculations arising in clustering and classifying datasets, and in the training of artificial neural networks, reduce to sampling a smooth probability distribution in high dimension and are frequently treated using the techniques of statistical physics [12,13]. In such systems, *a priori* knowledge of the CVs is typically not available, so methods that can automatically determine CVs are of high potential value.

Diffusion maps [14,15] provide a dimensionality reduction technique which yields a parametrized description of the underlying low-dimensional manifold by computing an approximation of a Fokker–Planck operator on the trajectory point-cloud sampled from a probability distribution (typically the Boltzmann–Gibbs distribution corresponding to prescribed temperature). The construction is based on a normalized graph Laplacian matrix. In an appropriate limit (discussed below), the matrix converges (point-wise) to the generator of

overdamped Langevin dynamics. The spectral decomposition of the diffusion map matrix thus yields an approximation of the continuous spectral problem on the point-cloud [16] and leads to natural CVs. Since the first appearance of diffusion maps [14], several improvements have been proposed including local scaling [17], variable bandwidth kernels [18] and target measure maps (TMDmap) [19]. The latter scheme extends diffusion maps on point-clouds obtained from a surrogate distribution, ideally one that is easier to sample from. Based on the idea of importance sampling, it can be used on biased trajectories, and improves the accuracy and application of diffusion maps in high dimensions [19]. Diffusion maps can be combined with the variational approach to conformation dynamics in order to improve the approximation of the eigenfunctions and provide eigenvalues that relate directly to physical relaxation timescales [20].

Diffusion maps underpin a number of algorithms that have been designed to learn the CV adaptively and thus enhance the dynamics in the learned slowest dynamics [21–24]. These methods are based on iterative procedures whereby diffusion maps are employed as a tool to gradually uncover the intrinsic geometry of the local states and drive the sampling toward unexplored domains of the state space, either through sequential restarting [24] or pushing [22] the trajectory from the border of the point-cloud in the direction given by the reduced coordinates. In [25], time structure-based independent component analysis was performed iteratively to identify the slowest degrees of freedom and to use metadynamics [8] to directly sample them. All these methods try to gather local information about the metastable states to drive global sampling, using ad hoc principles. In this paper, we provide a rigorous perspective on the construction of diffusion maps within a metastable state by formalizing the concept of a local equilibrium based on the *quasi-stationary distribution* (QSD) [26]. Moreover, we provide the analytic form of the operator which is obtained when metastable trajectories are used in computing diffusion maps.

Diffusion maps can also be used to compute committor functions [27], which play a central role in transition path theory [28]. The committor is a function which provides dynamical information about the connection between two metastable states and can thus be used as a reaction coordinate (importance sampling function for biasing or splitting methods, for example). Committors, or the 'commitment probabilities', were first introduced as 'splitting probability for ion-pair recombination' by Onsager [29] and appear, for example, as the definition of $p_{fold}$, the probability of protein folding [30,31]. Markov state models can in principle be used to compute committor probabilities [32], but high dimensionality makes grid-based methods intractable. The finite temperature string method [33] approximates the committor on a quasi-one-dimensional reaction tube, which is possible under the assumption that the transition paths lie in regions of small measure compared to the whole sampled state space. The advantage of diffusion maps is that the approximation of the Fokker–Planck operator holds on the whole space, and therefore we can compute the committor outside the reaction tube. Diffusion maps have already been used to approximate committor functions in [27,34]. In [27], in order to improve the approximation quality, a new method based on a point-cloud discretization for Fokker–Planck operators is introduced (however without any convergence result). A more recent work [34] uses diffusion maps for committor computations. Finally, we mention that artificial neural networks were used to solve for the committor in [35], although the approach is much different than that considered here.

The main conceptual novelty of this article lies in the insight on the local versus global perspective provided by the QSD. Thus, to be precise, compared to the work of [27], we clarify the procedure for going from the local perspective to designing an enhanced sampling method and we apply our methods to much more complicated systems (only toy models are treated in [27]). The second, more practical, contribution is the demonstration of the use of diffusion maps to identify the metastable states and to directly compute the committor function. We consider, for example, the use of the diffusion map as a diagnostic tool for transition out of a metastable state. A third contribution lies in drafting an enhanced sampling algorithm based on QSD and diffusion maps. A careful implementation and application for small biomolecules shows the relevance and potential of this methodology for practical applications. Algorithm 1 detailed in §5 will provide a starting point for further investigations. The current article serves to bridge works

in the mathematical and computational science literatures, thus helps to establish foundations for future rigorously based sampling frameworks.

This paper is organized as follows: in §2, we start with the mathematical description of overdamped Langevin dynamics and diffusion maps. In §3, we formalize the application of diffusion maps to a local state using the QSD. We present several examples illustrating the theoretical findings. In §4, we define committor probabilities and the diffusion map-based algorithm to compute them. We also apply our methodology to compute CVs, metastable states and committors for various molecules, including the alanine dipeptide and a deca-alanine system. In §5, we show how the QSD can be used to reveal transitions between molecular conformations. Finally, we conclude by taking up the question of how the QSD can be used as a tool for the enhanced sampling of large-scale molecular models, paving the way for a full implementation of the described methodology in software framework.

## 2. Langevin dynamics and diffusion maps

We begin with the mathematical description of overdamped Langevin dynamics, which is used to generate samples from the Boltzmann distribution. By introducing the generator of the Langevin process, we make a connection to the diffusion maps which is formalized in the following section. We review the construction of the original diffusion maps and define the target measure diffusion map, which removes some of its limitations.

### (a) Langevin dynamics and the Boltzmann distribution

We denote the configuration of the system by $x \in \mathcal{D}$, where, depending on the application, $\mathcal{D} = \mathbb{R}^d$, $\mathcal{D}$ is a subset of $\mathbb{R}^d$ or $\mathcal{D} = (\mathbb{R}/\mathbb{Z})^d$ for systems with periodic boundary conditions. Overdamped Langevin dynamics is defined by the stochastic differential equation

$$dx_t = -\nabla V(x_t)\, dt + \sqrt{\frac{2}{\beta}}\, dW_t, \tag{2.1}$$

where $W_t$ is a standard $d$-dimensional Wiener process, $\beta > 0$ is the inverse temperature and $V(x)$ is the potential energy driving the diffusion process. The *Boltzmann–Gibbs* measure is invariant under this dynamics:

$$\mu(dx) = Z^{-1} e^{-\beta V(x)}\, dx \quad \text{and} \quad Z = \int_{\mathcal{D}} e^{-\beta V(x)}\, dx. \tag{2.2}$$

The ergodicity property is characterized by the almost sure (a.s.) convergence of the trajectory average of a smooth observable $A$ to the average over the phase space with respect to a probability measure, in this case $\mu$:

$$\lim_{t \to \infty} \widehat{A}_t = \mathbb{E}_\mu(A) \quad \text{a.s.,} \quad \widehat{A}_t := \frac{1}{t} \int_0^t A(x_s)\, ds. \tag{2.3}$$

The infinitesimal generator of the Markov process $(x_t)_{t \geq 0}$, a solution of (2.1), is the differential operator

$$\mathcal{L}_\beta = -\nabla V \cdot \nabla + \beta^{-1} \Delta, \tag{2.4}$$

defined for example on the set of $C^\infty$ functions with compact support. The fact that $\mathcal{L}_\beta$ is the infinitesimal generator of $(x_t)_{t \geq 0}$ means that (e.g. [36]):

$$\forall x, \quad \frac{d}{dt} \Big[ \mathbb{E}(A(x_t) \mid x_0 = x) \Big] \Big|_{t=0} = \mathcal{L}_\beta A(x),$$

where $A$ is $C^\infty$ compactly supported function and $x_0 = x$ is the initial condition at time $t = 0$. Another way to make a link between the differential operator (2.4) and the stochastic differential

equation (2.1) is to consider the law of the process $x_t$. Let us denote by $\psi(t, x)$ the density of $x_t$ at time $t$. Then $\psi$ is a solution of the Fokker–Planck equation

$$\partial_t \psi = \mathcal{L}_\beta^* \psi \quad \text{and} \quad \psi(0) = \psi_0,$$

where $\psi_0$ is the density of $x_0$ and $\mathcal{L}_\beta^*$ is the $L^2$ adjoint of $\mathcal{L}_\beta$. Under the assumption on the smoothness of the potential and the compactness of the domain $\mathcal{D}$, the solution for positive time can be written as

$$\psi(t, x) = \sum_{k=0}^{\infty} c_k e^{\lambda_k t} \phi_k(x), \tag{2.5}$$

with eigenvalues $\{\lambda_k\}_{k=0}^{\infty}$ with $\lambda_0 = 0 > \lambda_1 \geq \lambda_2 \geq \dots$ and eigenfunctions $\{\phi_k\}_{k=0}^{\infty}$ of $\mathcal{L}_\beta^*$. The eigenfunctions are smooth functions and the sum (2.5) converges uniformly in $x$ for all times $t > t_0 > 0$ [15]. The ergodicity implies that $\psi(t, x) \to c_0 \phi_0(x)$ as $t \to \infty$, and therefore the first term $c_0 \phi_0$ in the sum (2.5) is equal to $\mu$. The convergence rate is determined by the next dominant eigenfunctions and eigenvalues. A $k$-dimensional diffusion map at time $t$ is a lower dimensional representation of the system defined as a nonlinear mapping of the state space to the Euclidean space with coordinates given by the first $k$ eigenfunctions:

$$G_k(t, x) := (e^{\lambda_1 t} \phi_1(x), \dots, e^{\lambda_k t} \phi_k(x)).$$

The diffusion distance is then the Euclidean distance between the diffusion map coordinates (DC).

Finally, we would like to stress that although the eigenfunctions of the kernel matrix do approximate point-wise the spatial eigenfunctions of the Fokker–Planck operator for the dynamics (2.1), there is not necessarily a relationship between the diffusion map matrix eigenvalues and the eigenvalues the eigenvalues of the dynamics used to generate the samples distributed according to $\mu$, which is typically the underdamped Langevin dynamics as shown in [20] (see also remark 2.1 below). This connection would be provided by constructing a diffusion map-based approximator of the generator of underdamped Langevin dynamics (with finite friction).

## (b) Diffusion maps

The diffusion map [14] reveals the geometric structure of a manifold $\mathcal{M}$ from given data $\mathbb{D}^{(m)}$ by constructing a $m \times m$ matrix that approximates a differential operator. The relevant geometric features of $\mathcal{M}$ are expressed in terms of the dominant eigenfunctions of this operator.

The construction requires that a set of points $\mathbb{D}^{(m)} := \{x_1, x_2, \dots, x_m\} \subset \mathbb{R}^N$ ($N > 0$) which have been sampled from a distribution $\pi(x)$ lie on a compact $d$-dimensional differentiable submanifold $\mathcal{M} \subset \mathbb{R}^N$ with dimension $d < N$. The original diffusion maps introduced in [14,37] are based on the isotropic kernel $h_\varepsilon(x, y) = h(\|x - y\|^2 / \varepsilon)$, where $h$ is an exponentially decaying function, $\varepsilon > 0$ is a scale parameter and $\|\cdot\|$ is a norm[1] in $\mathbb{R}^N$. A typical choice is

$$h_\varepsilon(x, y) = \exp\left(-(4\varepsilon)^{-1} \|x - y\|^2\right). \tag{2.6}$$

In the next step, an $m \times m$ kernel matrix $K_\varepsilon$ is built by the evaluation of $h_\varepsilon$ on the set $\mathbb{D}^{(m)}$. This matrix is then normalized several times to give a matrix $P_\varepsilon$ that can be interpreted as the generator of a Markov chain on the data. To be precise, the kernel matrix $K_\varepsilon$ is normalized using the power $\alpha \in [0, 1]$ of the estimate $q$ of the density $\pi$, usually obtained from the kernel density estimate $q_i = \sum_{j=1}^{N} K_{ij}$ as the row sum of $K_\varepsilon$. In some cases, the analytic expression of the density $\pi(x)$ is known and we can set directly $q(x) = \pi(x)$. After obtaining the transition matrix

$$P_\varepsilon = D_\alpha^{-1} K_\varepsilon, \tag{2.7}$$

where $D_\alpha = \text{diag}(q^{-\alpha})$, we compute in the last step the normalized graph Laplacian matrix

$$L_\varepsilon = \varepsilon^{-1}(P_\varepsilon - I). \tag{2.8}$$

[1]For example, Euclidean or RMSD, which is the most commonly used norm in molecular simulations [20].

As reviewed in [16], for given $\alpha$ and sufficiently smooth functions $f$, the matrix $L_\varepsilon$ converges in the limit $m \to \infty$ point-wise to an integral operator describing a random walk in continuous space and discrete time, which in the limit $\varepsilon \to 0$ converges to the infinitesimal generator of the diffusion process in continuous space and time. Using the notation $[f] = (f(x_1), \ldots, f(x_m))^\mathsf{T}$ for representing functions evaluated on the dataset $\mathbb{D}^{(m)}$ as vectors such that $[f]_i = f(x_i)$, we formally write the point-wise convergence: for $\alpha \in [0, 1]$, for $m \to \infty$ and $\varepsilon \to 0$,

$$(L_\varepsilon[f])_j \to \mathcal{L}f(x_j), \quad \text{for all } x_j \in \mathbb{D}^{(m)},$$

where $\mathcal{L}$ is the operator

$$\mathcal{L}f = \Delta f + (2 - 2\alpha)\nabla f \cdot \frac{\nabla \pi}{\pi}, \tag{2.9}$$

where $\Delta$ is the Laplace–Beltrami operator on $\mathcal{M}$ and $\nabla$ is the gradient operator on $\mathcal{M}$. Note that in the special case $\alpha = 1/2$, and for the choice $\pi = Z^{-1}e^{-\beta V}$ (Boltzmann–Gibbs), the approximated operator corresponds to the generator of overdamped Langevin dynamics (2.4), such that

$$\mathcal{L} = \beta \mathcal{L}_\beta. \tag{2.10}$$

For this reason, we focus on the choice $\alpha = 1/2$ throughout this work.

Consequently, if there are enough data points for accurate statistical sampling, eigenvectors of $L_\varepsilon$ approximate discretized eigenfunctions of $\mathcal{L}$. Then eigenvectors of $L_\varepsilon$ approximate solutions to the eigenproblem associated with $\mathcal{L} : L_\varepsilon[\psi] = \lambda[\psi]$, an approximation of

$$\mathcal{L}\psi(x) = \lambda \psi(x), \forall x \in \mathcal{M} \subset \mathrm{supp}(\pi).$$

The spectral decomposition of $L_\varepsilon$ provides real, non-positive eigenvalues $0 = \lambda_0 > \lambda_1 \geq \lambda_2 \geq \cdots \geq \lambda_m$ sorted in decreasing order. The dominant eigenfunctions allow for a structure preserving embedding $\Psi$ of $\mathbb{D}^{(m)}$ into a lower dimensional space and hence reveal the geometry of the data.

Singer [37] showed that for uniform density $\pi$, the approximation error for fixed $\varepsilon$ and $m$ is

$$(L_\varepsilon[f])_j = \mathcal{L}f(x_j) + O\left(\varepsilon, m^{-1/2}\varepsilon^{-1/2-d/4}\right). \tag{2.11}$$

**Remark 2.1 (Infinite friction limit).** In molecular dynamics, trajectories are usually obtained by discretizing [38] the (underdamped) Langevin dynamics:

$$\left. \begin{aligned} \mathrm{d}x_t &= M^{-1}p_t \, \mathrm{d}t \\ \mathrm{d}p_t &= -\nabla V(x_t) \, \mathrm{d}t - \gamma M^{-1}p_t \, \mathrm{d}t + \sqrt{\tfrac{2\gamma}{\beta}} \, \mathrm{d}W_t, \end{aligned} \right\} \tag{2.12}$$

where $x \in \mathbb{R}^d$ are positions, $p \in \mathbb{R}^d$ momenta, $W_t$ is a standard $d$-dimensional Wiener process, $\beta > 0$ is proportional to the inverse temperature, $M = \mathrm{diag}\,m_1, \ldots, m_d$ is the diagonal matrix of masses, and $\gamma > 0$ is the friction constant. The generator of this process is

$$\mathcal{L}_\gamma = M^{-1}p \cdot \nabla_x - \nabla V(x) \cdot \nabla_p + \gamma \left(-M^{-1}p \cdot \nabla_p + \frac{1}{\beta}\Delta_p\right).$$

Recall that the diffusion maps approximate the generator (2.9) which is the generator of the overdamped Langevin dynamics, an infinite-friction-limit dynamics of the Langevin dynamics (2.12). Diffusion maps therefore provide the dynamical information in the large friction limit $\gamma \to +\infty$, rescaling time as $\gamma t$, or in a small mass limit $(M \to 0)$.

**Remark 2.2.** Note that diffusion maps require the data to be distributed with respect to $\pi(x)\,\mathrm{d}x$, which appears in the limiting operator (2.9). It implies that even though diffusion maps eventually provide an approximation of the generator of overdamped Langevin dynamics (2.10), one can use trajectories from any ergodic discretization of underdamped Langevin dynamics to approximate the configurational marginal $\pi(x)\,\mathrm{d}x$ as, for example, the BAOAB integrator [39].

## (c) The target measure diffusion map

If $\pi$ is particularly difficult to sample, it might be desirable to use points which are not necessarily distributed with respect to $\pi$ in order to compute an approximation to the operator

$$\mathcal{L} = \nabla \log(\pi) \cdot \nabla + \Delta. \tag{2.13}$$

For this purpose, the target measure diffusion map was recently introduced [19]. The main advantage is that it allows construction of an approximation of $\mathcal{L}$ even if the data points are distributed with respect to some distribution $\mu$ such that supp $\pi \subset$ supp $\mu$. The main idea is to use the previously introduced kernel density estimator $q_\varepsilon$, already obtained as an average of the kernel matrix $K_\varepsilon$. Since we know the density of the target distribution $\pi$, the matrix normalization step can be done by re-weighting $q$ with respect to the target density, which allows for the matrix normalization in an importance sampling sense. More precisely, the target measure diffusion map (TMDmap) is constructed as follows: the construction begins with the $m \times m$ kernel matrix $K_\varepsilon$ with components $(K_\varepsilon)_{ij} = k_\varepsilon(x_i, x_j)$ and the kernel density estimate $q_\varepsilon(x_i) = \sum_{j=1}^{m}(K_\varepsilon)_{ij}$. Then the diagonal matrix $D_{\varepsilon,\pi}$ with components $(D_{\varepsilon,\pi})_{ii} = \pi^{1/2}(x_i)q_\varepsilon^{-1}(x_i)$ is formed and the kernel matrix is right normalized with $D_{\varepsilon,\pi}$:

$$K_{\varepsilon,\pi} = K_\varepsilon D_{\varepsilon,\pi}.$$

Let us define $\tilde{D}_{\varepsilon,\pi}$ as the diagonal matrix of row sums of $K_{\varepsilon,\pi}$, that is,

$$(\tilde{D}_{\varepsilon,\pi})_{ii} = (K_{\varepsilon,\pi}[\mathbf{1}])_i = \sum_{j=1}^{m}(K_{\varepsilon,\pi})_{ij}.$$

Finally, the TMDmap matrix is built as

$$L_{\varepsilon,\pi} = \varepsilon^{-1}\left(\tilde{D}_{\varepsilon,\pi}^{-1}K_{\varepsilon,\pi} - I\right). \tag{2.14}$$

In [19], it is shown that $L_{\varepsilon,\pi}$ converges point-wise to the operator (2.13).

**Remark 2.3 (Nyström extension of the eigenvectors [40]).** Note that the $j$th eigenvector $[\psi]$ of the matrix $L_\varepsilon$ can be extended on $x \in \mathcal{M}$ as

$$\psi_j(x) = \frac{1}{\lambda_j}\sum_{i=1}^{N}\frac{h_\varepsilon(x,x_i)}{D(x)}[\psi_j]_i,$$

where $\lambda_j \neq 0$ is the corresponding eigenvalue, $D(x) = \sum_{i=1}^{N} h_\varepsilon(x,x_i)$ and $[\psi_j]_i = \psi_j(x_i)$.

## (d) Dirichlet boundary problems

Diffusion maps provide a matrix $L_\varepsilon$, which converges point-wise to the generator $\mathcal{L}$ defined in (2.9). This method can be used to solve the following eigenvalue problem with homogeneous Dirichlet boundary conditions: find $(\lambda, f)$ such that $\mathcal{L}f = \lambda f$ in $\Omega, f = 0$ in $\partial\Omega$. In the following example, we solve a linear eigenvalue problem with Dirichlet boundary conditions. In the first step, we construct $L_\varepsilon$ as the diffusion map approximation of $\mathcal{L}$ on the point-cloud $\{x_i\}_{i=1}^{m}$. In order to express the Dirichlet boundary condition, we identify points outside the domain $\Omega$ that we define as $\mathcal{C} := \{x_j\}_{j\in J}$ where the set of indices $J := \{j : x_j \notin \Omega\}$. Finally, we solve the eigenvalue problem with matrix $L_\varepsilon$, in which rows with indices in $J$ have been set to zero.

Let us illustrate this on the following one-dimensional eigenvalue problem:

$$\mathcal{L}v = \lambda v, \quad v \in (-1, 1), \quad v = 0, v \in \{-1, 1\}, \tag{2.15}$$

with $\mathcal{L} = -x\partial + \partial^2$. The eigenfunctions are $\psi_k(x) = (1/2)\sin(\pi(x+1)(k+1))$. To approximate the solution of (2.15) using diffusion maps with $\alpha = 1/2$, we have generated $10^6$ points from a discretized overdamped Langevin trajectory with potential $V(x) = x^2/2$, using a second-order numerical scheme [39]. In figure 2a, we show the diffusion map approximation of the eigenfunctions $\psi_k/\|\psi_k\|_2$. From figure 2b, we observe that the decay of the mean absolute error of

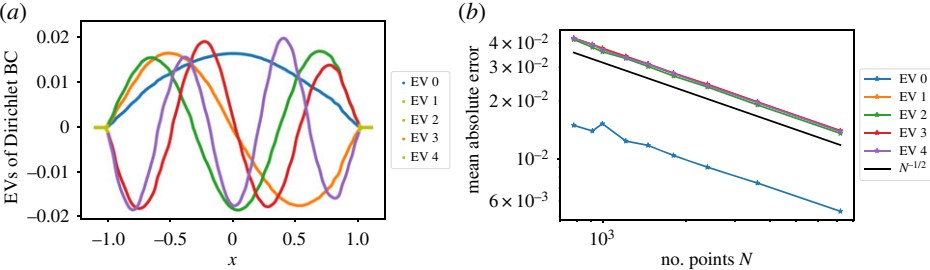

**Figure 2.** (*a*) Eigenvectors obtained from diffusion maps. (*b*) Mean absolute error on the first five eigenfunctions over the number of samples. (Online version in colour.)

the normalized eigenfunctions is asymptotically proportional to $N^{-1/2}$, where $N$ is the number of samples. Different numbers $N$ of samples were obtained by sub-sampling the trajectory.

# 3. Defining a 'local' perspective in diffusion-map analysis

We now concentrate on the case when diffusion maps are built using trajectories of the Langevin dynamics. As we have reviewed in the Introduction, many iterative methods aim at gradually uncovering the intrinsic geometry of the local states. The information obtained from the metastable states can then be used, for example, to accelerate the sampling towards unexplored domains of the state space.

The approximation error of diffusion maps (2.11) scales in $O(m^{-1/2})$, $m$ being the number of samples. In order to provide an approximation of a Fokker–Planck operator (2.4) using a point-cloud obtained from the process $x_t$, a solution of (a discretized version of) (2.1), the time averages should have converged with a sufficiently small statistical error. In this section, using the notion of the QSD, we explain to which operator converges a diffusion map approximation constructed on the samples in a metastable subset of the state space and why it is possible to obtain convergence of the approximation in this set-up.

## (a) Quasi-stationary distribution

The QSD is a local equilibrium macro-state describing a process trapped in a metastable state $\Omega \subset \mathbb{R}^d$ (e.g. [26]). The QSD $\nu$ can be defined as follows: for all smooth test functions $A : \mathbb{R}^d \to \mathbb{R}$,

$$\forall X_0 \in \Omega, \quad \lim_{t \to \infty} \mathbb{E}(A(X_t) \mid \tau > t) = \int_\Omega A \, d\nu,$$

where $\Omega$ is a smooth bounded domain in $\mathbb{R}^d$, which is the support of $\nu$ and the first exit time $\tau$ from $\Omega$ for $X_t$ is defined by

$$\tau = \inf\{t > 0 : X_t \notin \Omega\}.$$

The following mathematical properties of the QSD were proved in [41]. The probability distribution $\nu$ has a density $v$ with respect to the Boltzmann–Gibbs measure $\pi(dx) = Z^{-1} e^{-\beta V(x)} \, dx$. The density $v$ is the first eigenfunction of the infinitesimal generator $\mathcal{L}$ of the process $X_t$, with Dirichlet boundary conditions on $\partial\Omega$:

$$\begin{cases} \mathcal{L}v = -\lambda v, & \text{in } \Omega, \\ v = 0, & \text{on } \partial\Omega, \end{cases} \tag{3.1}$$

where $-\lambda < 0$ is the first eigenvalue. The density of $\nu$ with respect to the Lebesgue measure is thus

$$\forall x \in \Omega, \quad \nu(x) = \frac{v(x) e^{-\beta V(x)}}{\int_\Omega v(x) e^{-\beta V(x)} \, dx}. \tag{3.2}$$

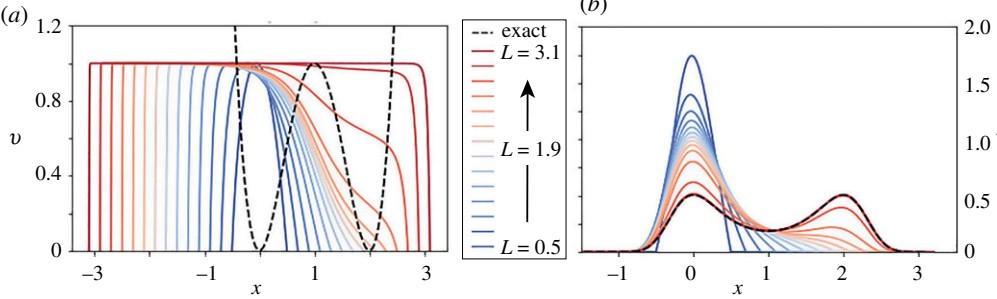

**Figure 3.** Quadratic potential in one dimension. (*a*) The approximation of the first eigenfunction $v$ of Dirichlet boundary eigenvalue problem (3.1). (*b*) The QSD $\nu$. (Online version in colour.)

Let us consider the situation when $\Omega$ is a metastable state for the dynamics $(X_t)$, and $X_0 \in \Omega$. Then, for a long time, the process remains in $\Omega$. If the diffusion map is built using those samples in $\Omega$, it then provides an approximation of the Kolmogorov operator (2.9) where $\pi$ is replaced by the QSD $\nu$, namely

$$\mathcal{L}_\Omega f := \Delta f + (2 - 2\alpha)\nabla f \cdot \frac{\nabla v}{v} = \Delta f + (2 - 2\alpha)\nabla f \cdot (\nabla \ln(v) - \beta \nabla V), \quad \text{on supp}(\Omega). \quad (3.3)$$

Notice that if $\Omega = \mathcal{D}$, $\nu = \pi$ and we recover the operator (2.9) with respect to the distribution $\pi$. In the case when $\Omega$ is in the basin of attraction of a local minimum $x_0$ of $V$ for the dynamics $\dot{x} = -\nabla V(x)$, the QSD with density $\nu$ defined by (3.2) is exponentially close to $\pi(x) = Z^{-1}e^{-\beta V(x)}\,\mathrm{d}x$ on any compact in $\Omega$: the two distributions differ essentially on the boundary $\partial\Omega$. More precisely, as proved in [42, (Lemma 23, Lemma 85)] for example (see also [43, (Theorem 3.2.3)]), for any compact subset $K$ of $\Omega$, there exists $c > 0$ such that, in the limit $\beta \to \infty$,

$$\left\| \frac{1_K \exp(-\beta V)}{\int_K \exp(-\beta V)} - \nu \right\|_{L^2(\Omega)} = \mathcal{O}(\exp(-\beta c)).$$

In order to illustrate these ideas, we compare the diffusion map constructed from a trajectory in a metastable state and points from a trajectory which has covered the whole support of the underlying distribution. As explained above, the distribution of the samples in the metastable state is the QSD and diffusion maps provide an approximation of the operator (3.3).

Let us next illustrate the QSD and explain how it differs from the stationary distribution on a simple one-dimensional example. The density of the QSD (3.2) can be obtained using accurate numerical approximation of the solution $v$ of the Dirichlet problem (3.1) by a pseudo-spectral Chebyschev method [44,45] in interval $[-L, L]$ with $L > 0$, with the grid chosen fine enough to provide sufficient numerical accuracy. We consider a simple double-well potential $V(x) = ((x - 1)^2 - 1)^2$. To illustrate the convergence of the QSD $\nu = Z_v^{-1}ve^{-V}$ to $\pi = Z^{-1}e^{-V}$ we increase the interval size by plotting the approximation for increasing values of $L$. In figure 3a, we plot the approximation of the solution $v$ of the Dirichlet eigenvalue problem (3.3) on $[-L, L]$ for several values of $L > 0$. As expected, we observe that $v(x) \to 1$ as $L \to \infty$. In figure 3b, we plot the corresponding quasi-stationary densities. As the size of the domain increases, the QSD converges to $\pi$.

In the next example, we sample from the Boltzmann distribution with a two-dimensional double-well [36, (Section 1.3.3.1)]:

$$V_{\mathrm{DW}}(x, y) = \tfrac{1}{6}(4(-x^2 - y^2 + w)^2 + 2h(x^2 - 2)^2 + ((x + y)^2 - w)^2 + ((x - y)^2 - w)^2), \quad (3.4)$$

with $h = 2$ and $w = 1$. We employ a second-order discretization of Langevin dynamics (2.1) at low temperature ($\beta = 10$). Due to this low temperature, the samples are trapped in the first well long enough to locally equilibrate to the QSD. We compute the statistical error of averages of various observables such as the configurational and the kinetic temperatures, because of the

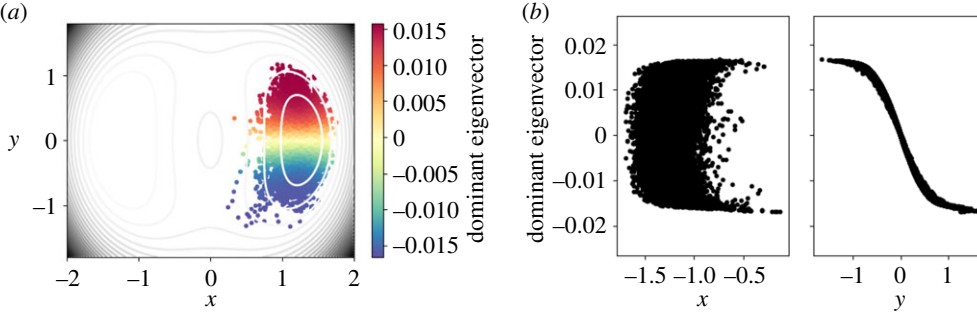

**Figure 4.** Local geometry. (*a*) The sampling of the metastable state. (*b*) The dominant eigenvector parametrizes the *y*-coordinate. (Online version in colour.)

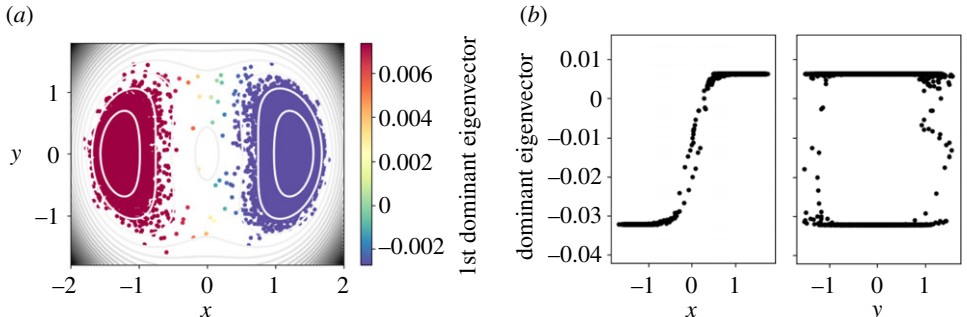

**Figure 5.** Global geometry. (*a*) The samples cover the whole support of the distribution. (*b*) The dominant eigenvector correlates with the *x*-coordinate. (Online version in colour.)

knowledge of the exact expected values of these observables.[2] We also track the first dominant eigenvalues from the diffusion map, in order to detect when the sampling has reached a local equilibrium. From this short trajectory, the diffusion map uncovers the geometry of the local state: the dominant eigenvector clearly parametrizes the *y*-coordinate, i.e. the slowest coordinate within the metastable state (figure 4). On the other hand, when the trajectory explores also the second well and hence covers the support of the distribution, the diffusion map parametrizes *x* as the slowest coordinate (figure 5).

These examples demonstrate that diffusion maps can be constructed using points of the QSD to uncover the slowest local modes. As we will show in §5 this property will eventually allow us to define local CVs which can guide the construction of sampling paths that exit the metastable state.

## 4. Global perspective: identification of metastable states and committors

We illustrate in this section how diffusion maps applied to global sampling can be used to approximate the generator of Langevin dynamics and committor probabilities between metastable states in infinite-friction limit.[3] We also use diffusion maps to automatically identify metastable sets in high-dimensional systems.

The committor function is the central object of transition path theory [28,47]. It provides a dynamical reaction coordinate between two metastable states $A \subset \mathcal{D}$ and $B \subset \mathcal{D}$. The committor

---

[2]The following equalities hold, respectively, for the kinetic and configurational temperatures: $k_B T = \mathbb{E}_\pi [p \cdot \nabla U(p)] = (\mathbb{E}_\pi [|\nabla V(x)|^2])/(\mathbb{E}[\Delta V(x)])$, where $k_B$ is the Boltzmann constant, $T$ is temperature and $U(p) = 1/2 p^\wedge T M^{-1} p$ is kinetic temperature.

[3]We are aware that diffusion maps do not provide access to dynamical properties. However, having access to a better reaction coordinate such as the committor can be used to obtain dynamical properties (for example in combination with Adaptive Multilevel Splitting [46]).

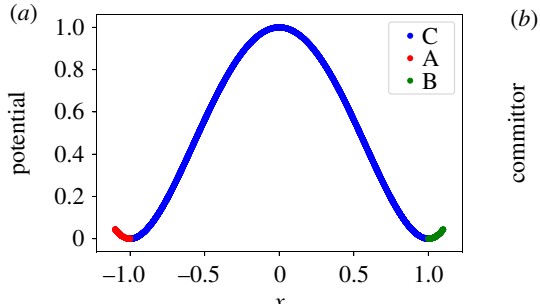 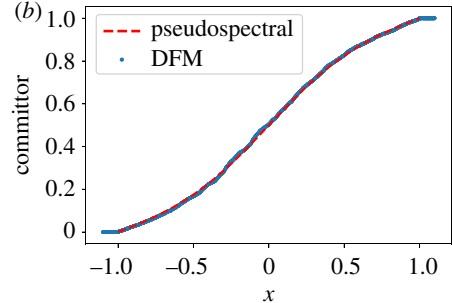

**Figure 6.** Double-well in one dimension. (*a*) A plot of the potential showing the three sets *A*, *B* and $C = \mathcal{D} \setminus (A \cup B)$. (*b*) The committor function approximation by diffusion maps (DFM) and an accurate pseudospectral method. (Online version in colour.)

function is the probability that the trajectory starting from $x \notin A \cup B$ reaches first *B* rather than *A*,

$$q(x) = \mathbb{P}(\tau_B < \tau_A \mid x_0 = x).$$

The committor *q* is also the solution of the following system of equations:

$$\begin{cases} \mathcal{L}q = 0, \text{in } \mathcal{D} \setminus (A \cup B), \\ q = 0, \text{in } A, \\ q = 1, \text{in } B. \end{cases} \tag{4.1}$$

From the committor function, one can compute the reaction rates, density and current of transition paths [48].

In the spirit of [27], we use diffusion maps to compute committors from the trajectory data. Given a point-cloud on $\mathcal{D}$, diffusion maps provide an approximation of the operator $\mathcal{L}$ and can be used to compute *q*. After computing the graph Laplacian matrix (2.8) (choosing again $\alpha = 1/2$), we solve the following linear system, which is a discretization of (4.1):

$$L_\varepsilon[c, c]q[c] = -L_\varepsilon[c, b]q[b], \tag{4.2}$$

where we defined by *c* and *b* indices of points belonging to the set $C = \mathcal{D} \setminus (A \cup B)$ and *B*, respectively, and $L_\varepsilon[I, J]$ the projection of the matrix or vector on the set of indices $I = \{I_k\}_{k=1}^{\#I}$ and $J = \{J_k\}_{k=1}^{\#J}$.

First, we compute the committor function for a one-dimensional double-well potential $V(x) = (x^2 - 1)^2$. We use a second-order discretized Langevin scheme with step size $\Delta t = 0.1$ to generate $10^5$ points and compute the TMDmap with $\epsilon = 0.1$. We fix the sets $A = [-1.1, -1]$ and $B = [1, 1.1]$ (figure 6*a*). In figure 6*b*, we compare the committor approximation with a solution obtained with a pseudo-spectral method [45], which would be computationally too expensive in high dimensions.

## (a) Algorithmic identification of metastable subsets

The most commonly used method for identifying metastable subsests is the Perron-cluster cluster analysis (PCCA), which exploits the structure of the eigenvectors [49–52]. In this work, we use the eigenvectors provided by diffusion maps in order to automatically identify the metastable subsets. The main idea is to compute the dominant eigenvectors of the transfer operator *P*, which are those with eigenvalues close to 1, excluding the first eigenvalue. We approximate *P* by $P_\varepsilon$ defined in (2.7). The metastable states can be clustered according to the 'sign' structure of the first dominant eigenvector, which is moreover constant on these states. More precisely, we find the maximal and the minimal points of the first eigenvector, which define the centres of the two sets. In the next step, we 'grow' the sets by including points with Euclidean distance in diffusion space

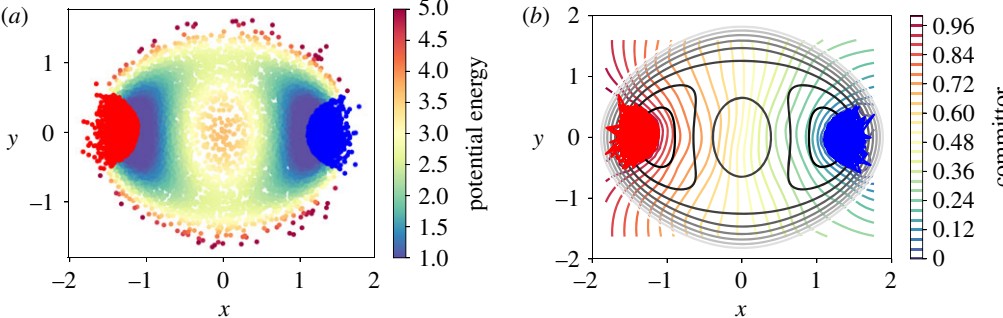

**Figure 7.** Two-dimensional double-well potential: committor approximations on the automatically identified metastable sets. (*a*) The sampled trajectory with the automatically chosen sets set *A* (blue) and *B* (red). (*b*) The committor extended on a grid. Note that the value of 0.5 is close to *x* = 0 as expected. (Online version in colour.)

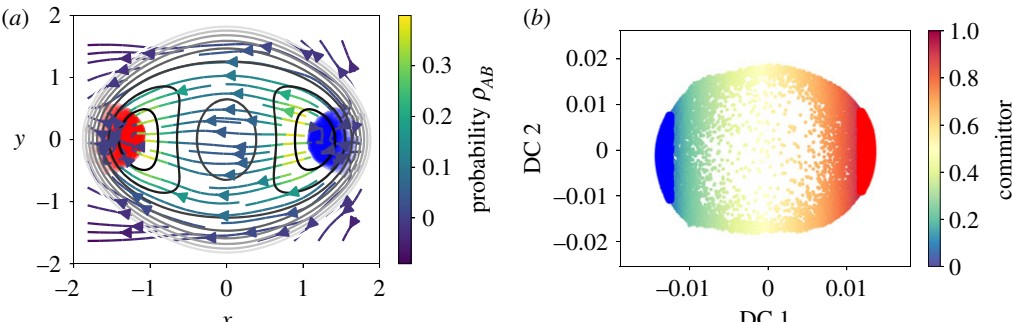

**Figure 8.** (*a*) The flux streamlines coloured by probability $\rho_{AB}$. (*b*) The diffusion map embedding and the chosen sets *A*, *B* (coloured blue and red). (Online version in colour.)

smaller than a fixed threshold. See, for example, figure 5*b* for an illustration in the case of the two-dimensional double-well potential.

In the second example, we consider a two-dimensional problem with potential (3.4) with $h = 2$ and $w = 1$. We generate $m = 10^5$ samples using discretized Langevin dynamics with timestep $\Delta t = 0.1$. We use $10^4$ points obtained by sub-sampling of the trajectory for the diffusion maps, which is chosen with kernel (2.6) and $\varepsilon = 0.1$. We compute the first two dominant eigenvectors and define the metastable subsets $A$ and $B$ using the first dominant eigenvector as described earlier.[4] In figure 7*a*, we see the sampling and the chosen sets. We compute the committor by solving the linear system (4.2) and extrapolate linearly the solution on a two-dimensional grid of $x$ and $y$; this is illustrated in the right panel of figure 7*b*. The representation of the committor in the diffusion coordinates is depicted in figure 8*b*.

## (b) High-dimensional systems

We now use diffusion maps to compute dominant eigenfunctions of the transition operator, identify metastable sets and approximate the committors of small molecules: alanine dipeptide and deca-alanine. We discuss the local and global perspective by computing diffusion maps inside the metastable states. For the example of deca-alanine, we use a trajectory from biased dynamics and use the TMDmap to approximate committor, comparing the dynamics at various temperatures.

[4]The results suggest that the method is robust with respect to the variation of the definition domains, under the assumption that the domains $A$ and $B$ are within the metastable state.

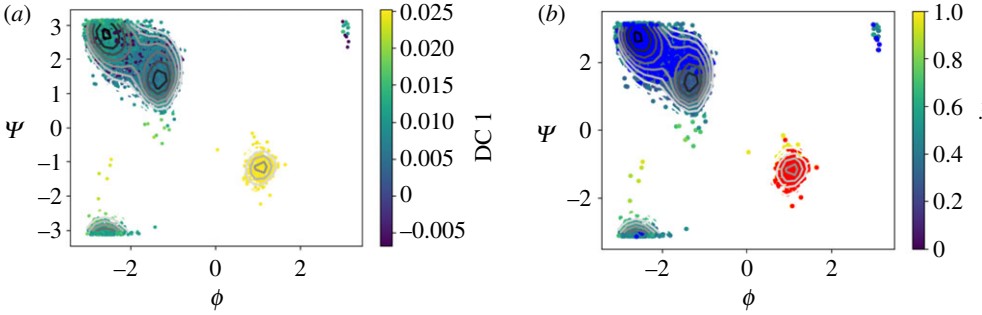

**Figure 9.** Alanine dipeptide. (*a*) The two metastable states uncovered by the first diffusion coordinate. (*b*) The committor function over sets A (red) and B (blue). The first eigenvector has a very high correlation with the committor function. (Online version in colour.)

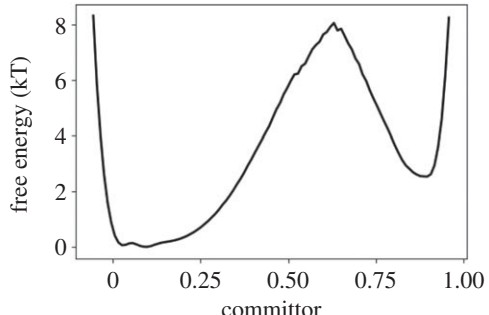

**Figure 10.** Extrapolated free energy profile along the committor function for alanine dipeptide.

### (i) Alanine dipeptide in vacuum

Alanine dipeptide ($CH_3$–CO–NH–$C_\alpha H CH_3$–CO–NH–$CH_3$) is commonly used as a toy problem for sampling studies since it has two well-defined metastable states which can be parametrized by the dihedral angles $\phi$ between C, N, $C_\alpha$, C and $\psi$ defined between N, $C_\alpha$, C, N.

We simulate a 20 ns Langevin dynamics trajectory (using the BAOAB integrator [38]) at temperature 300 K with a 2 fs stepsize, friction $\gamma = 1\,ps^{-1}$ and periodic boundary conditions using the openmmtools (OpenMMTools) library where the alanine dipeptide in vacuum is provided by the AMBER ff96 force field. We sub-sample and RMSD-align the configurations with respect to a reference one leaving only $5 \times 10^4$ points for the diffusion map analysis. We use kernel (2.6) with $\varepsilon = 1$ and the Euclidean metric.

In figure 9*a*, we show the first diffusion coordinate, which has opposite signs on the two metastable states. Since the transition is very rare, there are only very few points in the vicinity of the saddle point, the dominant eigenvector, however, clearly parametrizes the dihedral angles, and separates the two metastable sets. In order to define the reactive sets, we use the first diffusion coordinate as described in §4. Figure 9*a* shows the dominant diffusion coordinate, whereas the figure 9*b* shows the committor function. We observe that the committor strongly correlates with the first eigenvector, which is expected due to the definition of the metastable state given by the dominant eigenvector. Figure 10 shows the free energy profile of the committor function, which was extended from the sub-sampled points to the trajectory of length 20 ns using nonlinear regression (a multi-layer perceptron).[5]

---

[5]More precisely, the subset served as a training set and fitted model was evaluated on the full trajectory. We have performed cross-validation to find the right parameters of the neural network.

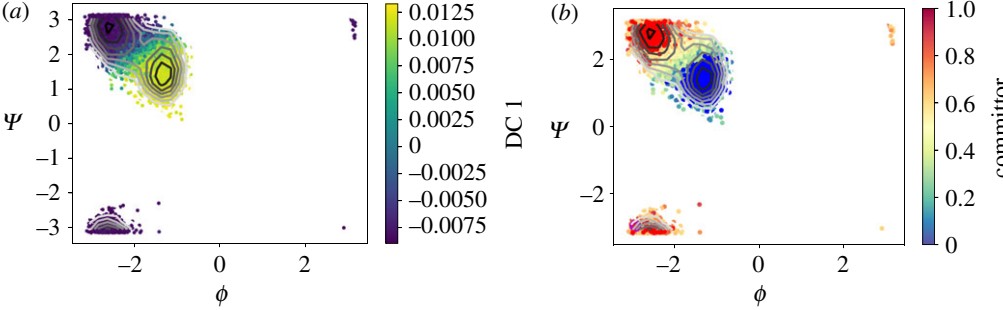

**Figure 11.** The first metastable state of alanine dipeptide. (*a*) The dominant eigenvector parametrizes the two wells of the first metastable state. (*b*) The committor approximation with sets *A*, *B* in blue and red. (Online version in colour.)

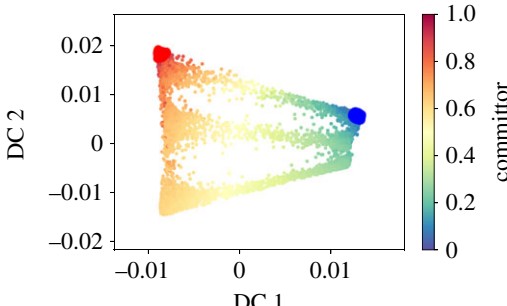

**Figure 12.** The embedding in the first two diffusion coordinates, coloured by the committor values. The sets *A*, *B* are shown in blue and red, respectively. (Online version in colour.)

In the previous example, we have used a globally converged trajectory. The diffusion map analysis therefore describes the dynamics from the global perspective. In order to illustrate the local perspective, we compute the diffusion maps and the committor approximation from a trajectory which has not left the first metastable state. The first eigenvector parametrizes the two wells of this metastable state and we define the reactive states as before using the first DC. The approximated committor assigns the probability 0.5 correctly to the saddle point between the two wells. Figure 11*a* shows the dominant eigenvector, figure 11*b* depicts the committor approximation with the automatically chosen sets. Figure 12 shows the corresponding diffusion map embedding coloured with respect to the committor values. Note that the diffusion maps used on the samples from this metastable state, whose distribution is the QSD, correlate with the dihedral angles. This observation suggests that alanine dipeptide in vacuum is a trivial example for testing enhanced dynamics methods using CVs learned on-the-fly because it is likely that the slow dynamics of the metastable state are similar to the global slow dynamics, which might not be the case in more complicated molecules.

## (ii) Deca-alanine

In the following example, we use a long trajectory of the deca-alanine molecule. This molecule has 132 atoms and a very complex free-energy landscape. A kinetic analysis was done in [53] showing the force-field-dependent slowest dynamics processes. A representation of the molecular conformations associated with the two main metastable states are shown in figure 1. The system is highly metastable: a standard simulation at 300 K requires at least 5 μs to converge [53]. Therefore, the trajectory studied here was obtained using the infinite-swap simulated tempering (ISST) [54] method with temperatures in the range from 300 K to 500 K (nominal stepsize 2 fs and nominal

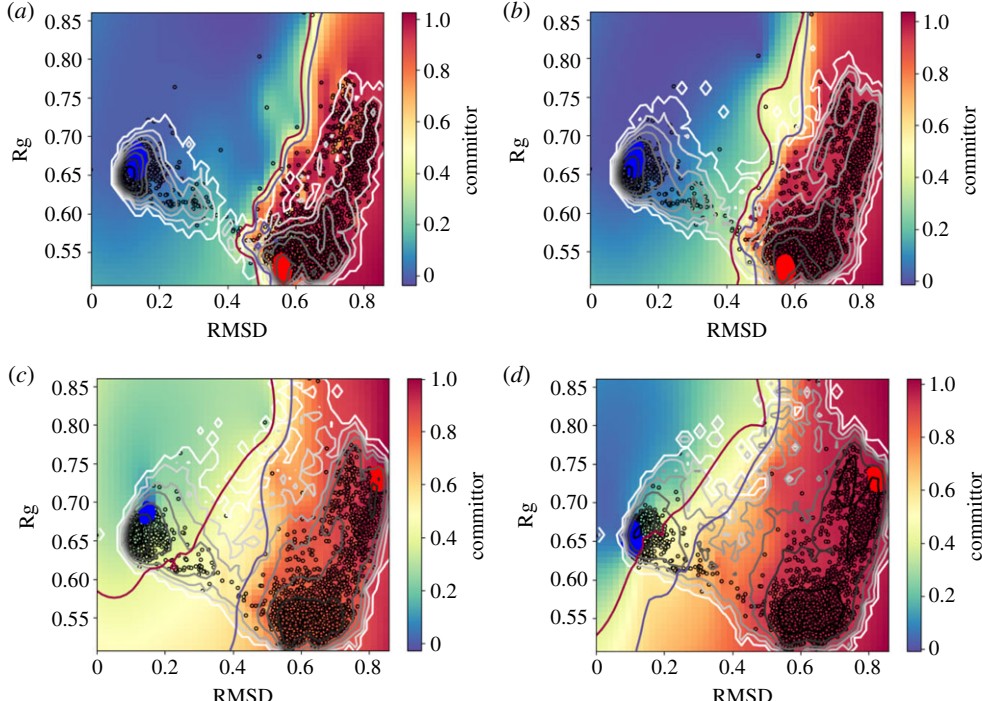

**Figure 13.** Committor of deca-alanine. Sets *A* (blue) and *B1* (red, top) and *B2* (red, bottom) were obtained from the first DC. The transition region is shown by committor isosurface lines at [0.4, 0.6]. The grey scale contours show the free-energies. Note that the slowest dynamics is different between low temperatures (300 K and 354 K) and high temperatures (413 K and 485 K). (*a*) 300 K, (*b*) 354 K, (*c*) 413 K and (*d*) 485 K. (Online version in colour.)

simulation length 2 μs). This method is incorporated into the MIST [55] library which is coupled to the GROMACS and Amber96 forcefields. ISST provides the weights which are necessary to recover the Boltzmann distribution at a given temperature. We use the weights within TMDmap allowing us to efficiently compute the committor iso-surfaces at various temperatures. We extend the committors in the root-mean-square deviation (RMSD) and the radius of gyration (Rg) for better visualization in figure 13 (however, note that the diffusion maps were applied to all degrees of freedom). We automatically identify the metastable sets using the dominant eigenvector. Note that at lower temperatures of 300 K and 354 K, the two states are *A* and *B*1, while at higher temperatures they are *A* and *B*2. See figure 1 for the corresponding conformations. The reason is that at the lower temperature, the dominant barrier is enthalpic and at higher temperatures, it is rather entropic, suggesting that the slowest transition is between the states *A* and *B*2, a state which is not very probable at low temperatures. The different dynamical behaviours can be also seen by the varying 0.5-committor probability shown in figure 13. The same result is also shown in the space of the first two diffusion map coordinates in figure 14.

## 5. From local to global: defining metastable states and enhanced sampling

In this section, we first illustrate how the spectrum computed from diffusion maps converges within the QSD. Next, we use diffusion coordinates built from samples within the metastable state to identify local CVs, which are physically interpretable and valid over the whole state space. Finally, we demonstrate that when used in combination with an enhanced dynamics such as metadynamics, these CVs lead to more efficient global exploration. The combination of these

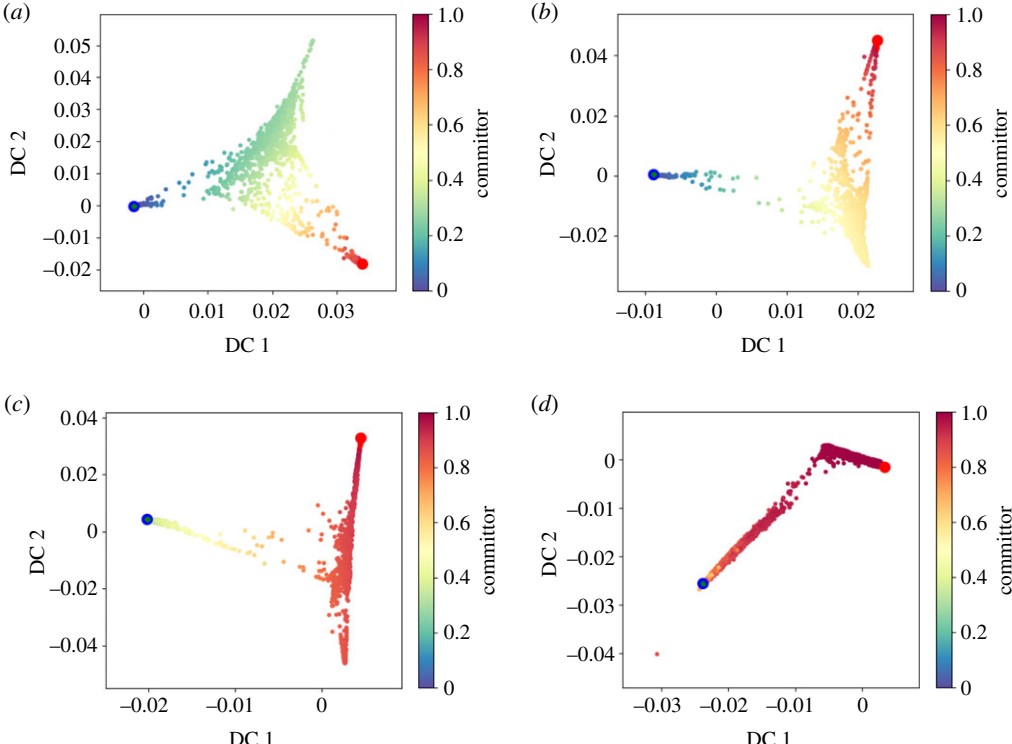

**Figure 14.** Committor of deca-alanine in diffusion coordinates. (*a*) 300 K, (*b*) 354 K, (*c*) 413 K and (*d*) 485 K. (Online version in colour.)

three procedures defines an algorithm for enhanced sampling which we formalize and briefly illustrate.

## (a) On-the-fly identification of metastable states

The local perspective introduced in §3 allows us to define a metastable state as an ensemble of configurations (snapshots) along a trajectory, for which the diffusion map spectrum converges. The idea is that, when trapped in a metastable state, one can compute the spectrum of the infinitesimal generator associated with the QSD, which will typically change when going to a new metastable state.

To illustrate this, we analyse an alanine dipeptide trajectory. Every 4000 steps, we compute the first dominant eigenvalues of the diffusion map matrix $L_\varepsilon$ (with $\alpha = 1/2$) by sampling 2000 points from the trajectory $\{x_n\}_{n=0}^m$ until the end of the last iteration $\tau = m\Delta t$. We observe in figure 15*a*, that the sampling has locally equilibrated during the first three iterations within the metastable state, and the eigenvalues have converged. The transition occurs after the fourth iteration, which we can see in figure 15*c* which shows the values of the dihedral angle $\phi$ during the simulation step. In figure 15*a*, we clearly observe a change in the spectrum at this point, with an increase in the spectral gap. After the trajectory has exited the metastable state, the eigenvalues begin evolving to new values, corresponding to the spectrum of the operator on the whole domain. The change in the spectrum allows us to detect the exit from the metastable state. Instead of tracking each of the first eigenvalues separately, we compute the average and the maximal difference of the dominant eigenvalues, the values of which we plot in figure 15*b*.

This example illustrates the local perspective on the diffusion maps: their application on a partially explored distribution provides an approximation of a different operator since the used

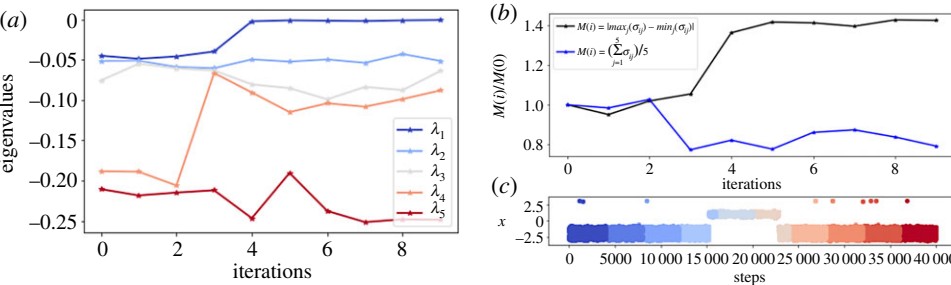

**Figure 15.** (a,b) The first dominant eigenvalues of matrix $L_\varepsilon$ with fixed number of points from alanine dipeptide trajectory. Plot (c) shows the values of $\phi$-angle over the simulation steps, with various colours corresponding to the iterations after which the spectrum is recomputed. Above, we plot the two functions of the dominant eigenvalues, computed over each iteration, which indicate the exit from the state as the eigenvalues suddenly change. Before the first transition to the second state, the spectrum is the one of $\mathcal{L}_\Omega$ (see (3.3) with $\alpha = (1/2)$), where $\Omega$ is the metastable state where the process is trapped. After the transition, the spectrum is evolving towards the spectrum of the operator $\mathcal{L}_{\mathbb{R}^d} = -\beta \nabla V + \Delta$ on the whole domain. (Online version in colour.)

samples are distributed with respect to a QSD. It is therefore possible to use samples from the local equilibrium to learn the slowest dynamics within the metastable state.

This observation can be used together with biasing techniques to improve sampling.

## (b) Enhanced sampling procedure for complex molecular systems

We now describe an outline of an algorithm for enhanced sampling.

Sampling from the QSD allows us to build high-quality local CVs (within the metastable state) by looking for the most correlated physical CVs to the DCs. In typical practice, the CVs are chosen from a list of physically relevant candidates, as, for example, the backbone dihedral angles or atom–atom contacts. There are several advantages to using physical coordinates instead of the abstract DC. First, the artificial DCs are only defined on the visited states, i.e. inside the metastable state. Extrapolated DCs outside the visited state lose their validity further from points used for the computation. By contrast, the physical coordinates can typically be defined over the entire state space. Second, the slowest physical coordinates might provide more understanding of the metastable state.

Once the best local CVs have been identified, we can use metadynamics to enhance the sampling, effectively driving the dynamics to exit the metastable state. In the next iteration, we suggest to use TMDmap to unbias the influence of metadynamics on the newly generated trajectory.

The above strategy can be summarized in the following algorithm:

**Algorithm 1.** Enhanced sampling algorithm based on local-global analysis.

> 1 Run molecular dynamics until the spectrum is converged (the reference walker is trapped);
> 2 *[Optional]* Refine the convergence of the spectrum and the DC by using a Fleming–Viot [56] process within the metastable state (each replica being initialized with the whole initial trajectory of the reference dynamics, for the diffusion map computation);
> 3 Identify CVs which are the most correlated with DCs;
> 4 Build an effective bias using, for example, metadynamics based on these CVs;
> 5 When an exit is observed for the reference walker (through a change in the spectrum), restart the procedure from step 1, keeping in memory the bias built using TMDmap;

As proof of concept, we will illustrate this method in a simplified setup based on metadynamics in the case of the alanine dipeptide. We stress that the construction of sampling

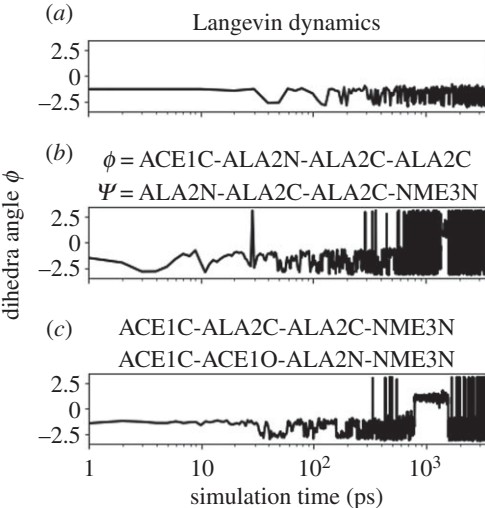

**Figure 16.** Alanine dipeptide: metadynamics with adaptive-CV (*c*) angles outperforms Langevin dynamics (*a*). Metadynamics with $\phi$, $\psi$ angles (*b*) is also more efficient than standard Langevin.

procedures for general macromolecules is a complex undertaking and many challenges still need to be addressed to implement the full proposed methodology for larger systems; this is the subject of current study by the authors.

We first run a short trajectory to obtain samples from the first metastable state.[6] This sampling is done by the Fleming–Viot process [56] with boundary defined by the converged spectrum which we regularly recompute during the sampling.[7] We compute the two dominant DCs and use them to select the two most correlated from a list of physical coordinates (a similar approach was presented in [57]). In this case, we looked at the collection of all possible dihedral angles from all the atoms except the hydrogens. In order to select the two most correlated CVs, we employed the Pearson correlation coefficient,

$$\rho(X, Y) = \frac{\mathrm{cov}(X, Y)}{\sigma_X \sigma_Y},$$

where $\sigma_Z$ is the standard deviation of a random variable $Z$. We found dihedral angles ACE1C-ALA2C-ALA2C-NME3N and ACE1C-ACE1O-ALA2N-NME3N to be the best candidates with highest correlations with the first and the second eigenvector, respectively. Note however, that there were also several other CVs with high correlations, among these the $\phi$ and $\psi$ angles. In the next step, we use the most correlated physical CVs within metadynamics and track the $\phi$-angle as a function of time. Moreover, we run metadynamics with *a priori* chosen CV's based on expert knowledge, specifically the $\phi$, $\psi$ angles. As shown in figure 16, there is no transition for the standard dynamics (*a*). On the other hand, metadynamics with the learned CVs exhibits several transitions, see figure 16*c*, similar to the knowledge-based CVs (*b*). We have also compared the least correlated CVs for metadynamics, and this approach was much worse than Langevin dynamics: we did not observe any exit during the expected Langevin exit time. These numerical experiments are strongly suggestive of the relevance of the most correlated CVs with the dominant DCs.[8] In conclusion, learning the slowest CVs from a local state provides important

[6]The 'first' state corresponds to the left top double well in figure 9.

[7]Alternatively, one can define the boundary by the free-energy in $\phi$, $\psi$ angles. We also found that resampling using Fleming–Viot improves the quality of diffusion maps.

[8]In this example, we did not compute the full expected exit times since the difference was significant for several realizations. To estimate the actual speed up of the sampling method, one would need to run thousands of trajectories and perform a detailed statistical analysis on the results, which is proposed for future study.

information allowing to escape the metastable state and hence enhance the sampling. Once the process leaves the first visited metastable state, one keeps the bias, new CVs are computed to update the bias once trapped elsewhere. This process can be iterated, and the weights from metadynamics can subsequently be unbiased by TMDmap to compute the relevant physical coordinates at every iteration.

We have thus demonstrated, at least in this specific case, how the local perspective can be used together with biasing techniques (metadynamics) to get more rapid sampling of the target distribution.

# 6. Conclusion and future work

Diffusion maps are an effective tool for uncovering a natural CV as needed for enhanced sampling. In this work, we have formalized the use of diffusion maps within a metastable state, which provides insight into diffusion map-driven sampling based on iterative procedures. The main theoretical tool for stating an analytical form of the approximated operator is the QSD. This local equilibrium guarantees the convergence of the diffusion map within the metastable state. We have also demonstrated that diffusion maps especially the TMDmap, can be used for committor computations in high dimensions. The low computational complexity aids in the analysis of molecular trajectories and helps to unravel the dynamical behaviour at various temperatures.

We have used the local perspective to identify the metastable state as a collection of states for which the spectrum computed by diffusion maps converges. We use the diffusion map eigenfunctions to learn physical coordinates which correspond to the slowest modes of the metastable state. This information not only helps to understand the metastable state, but leads to iterative procedure which can enhance the sampling.

Following the encouraging results we obtained in the last section, other techniques can be explored to fuel the iterative diffusion map sampling: for example, the adaptive biasing force method [58], metadynamics or dynamics biasing techniques as adaptive multilevel splitting [46], Forward Flux Sampling [59]. In the case of AMS, committor can be used as the one-dimensional reaction coordinate. It will be worth exploring these strategies for more complex molecules.

Finally, as a future work, we point out that the definition of the metastable states using the spectrum computed by diffusion map could be used within an accelerated dynamics algorithm, namely the parallel replica algorithm [60,61]. Specifically, one could use the Flemming–Viot particle process within the state (Step 2 of algorithm 1) to estimate the correlation time, using the Gelman–Rubin convergence diagnostic [56], restarting the sampling whenever the reference walker leaves prior to convergence and otherwise using trajectories generated from the FV process to compute exit times (using the diffusion map spectra to identify exit).

Data accessibility. Data are accessible as 'Martinsson & Trstanova [62]. The following libraries were used:—pydiffmap https://github.com/DiffusionMapsAcademics/pyDiffMap---OpenMM Eastman, Peter, *et al.* [63].
Authors' contributions. All authors contributed equally to the conception, design and interpretation of the presented work. More specifically, the project was initiated by B.L. and Z.T. Z.T. performed the theoretical analysis, designed and implemented the numerical simulations. T.L. contributed to the conception and presentation of the mathematical analysis, and to the numerical applications. All three authors shared in the paper writing and interpretation of numerical experiments contained in the article. All authors agreed to be accountable for all aspects of the work in ensuring that questions related to the accuracy or integrity of any part of the work are appropriately investigated and resolved.
Competing interests. The authors have no competing interests.
Funding. B.L and Z.T. were supported by EPSRC grant no. EP/P006175/1. B.L. was further supported by the Alan Turing Institute (EPSRC EP/N510129/1) as a Turing Fellow. T.L. is supported by the European Research Council under the European Union's Seventh Framework Programme (FP/2007-2013)/ERC grant agreement no. 614492.
Acknowledgements. The authors thank Ben Goddard and Antonia Mey (both at University of Edinburgh) for helpful discussions and the anonymous referees for useful criticism.

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
