## [Reviewer comments · Proceedings. Mathematical, Physical, and Engineering Sciences]

Review History

RSPA-2019-0036.R0 (Original submission)

Review form: Referee 1

Is the manuscript an original and important contribution to its field?

No

Is the paper of sufficient general interest?

Yes

Is the overall quality of the paper suitable?

Yes

Quality of the paper

A paper that may be acceptable after major revision.

Can the paper be shortened without overall detriment to the main message?

No

Do you think some of the material would be more appropriate as an electronic appendix?

No

For papers with colour figures – is colour essential?

Yes

If there is supplementary material, is this adequate and clear?

Not applicable

Are there details of how to obtain materials and data, including any restrictions that may apply?

Not applicable

Do you have any ethical concerns with this paper?

No

Recommendation?

Major revision is needed (please make suggestions in comments)

Comments to the Author(s)

See review attached

Review form: Referee 2

Is the manuscript an original and important contribution to its field?

Yes

Is the paper of sufficient general interest?

Yes

Is the overall quality of the paper suitable?

Yes

Quality of the paper

An excellent paper making an important contribution to the field: should be published.

Can the paper be shortened without overall detriment to the main message?

No

Do you think some of the material would be more appropriate as an electronic appendix?

No

For papers with colour figures – is colour essential?

Yes

If there is supplementary material, is this adequate and clear?

Not applicable

Are there details of how to obtain materials and data, including any restrictions that may apply?

Not applicable

Do you have any ethical concerns with this paper?

No

Recommendation?

Accept as is

Comments to the Author(s)

Authors of the manuscript with ID RSPA-2019-0036 have proposed an approach based on diffusion maps for identifying the metastable sets and the corresponding committor functions (in line with previous similar attempts from the literature such as the herein refs 9 and 40). The approach is well discussed and convincing examples are presented (both toy models and more complex cases). This Referee thinks that the works makes a very good contribution to the field and deserves publication as is.

Review form: Referee 3

Is the manuscript an original and important contribution to its field?

Yes

Is the paper of sufficient general interest?

Yes

Is the overall quality of the paper suitable?

Yes

Quality of the paper

A paper that may be acceptable after major revision.

Can the paper be shortened without overall detriment to the main message?

No

Do you think some of the material would be more appropriate as an electronic appendix?

No

For papers with colour figures - is colour essential?

Yes

If there is supplementary material, is this adequate and clear?

Not applicable

Are there details of how to obtain materials and data, including any restrictions that may apply?

Yes

Do you have any ethical concerns with this paper?

No

Recommendation?

Major revision is needed (please make suggestions in comments)

Comments to the Author(s)

See attached file

Review form: Referee 4

Is the manuscript an original and important contribution to its field?

No

Is the paper of sufficient general interest?

No

Is the overall quality of the paper suitable?

Yes

Quality of the paper

A paper that is of insufficient interest, quality or importance.

Can the paper be shortened without overall detriment to the main message?

Yes

Do you think some of the material would be more appropriate as an electronic appendix?

Yes

For papers with colour figures – is colour essential?

Yes

If there is supplementary material, is this adequate and clear?

Not applicable

Are there details of how to obtain materials and data, including any restrictions that may apply?

Not applicable

Do you have any ethical concerns with this paper?

No

Recommendation?

Reject – article is scientifically unsound

Comments to the Author(s)

The manuscript by Trstanova and coworkers is about Diffusion Map, a dimensionality reduction technique first introduced by Coifman and Lafon in 2006, intended to identify meaningful, i.e., slowly-varying collective variables of large complex systems. Diffusion map was originally shown to approximate the spatial eigenvectors of the Fokker-Planck equation that describes the dynamics of brownian systems such as proteins and other molecular systems in a water environment. As mentioned by the authors in the introduction, an important application of Diffusion Map is the construction of adaptive sampling methods which are based on the biasing of molecular dynamics towards meaningful coordinates – in the present case, the diffusion coordinates – which enable the exploration of the conformational space in a more efficient way.

Such methods are, in my opinion, crucial as standard molecular dynamics performed on regular CPUs are unable to explore relevant transitions or conformational changes of realistic protein systems in a reasonable amount of time.

In the current manuscript, Trstanova and coworkers explore different aspects of diffusion maps (and TM diffusion maps) including mathematical aspects (section 2), properties of diffusion maps when constructed upon global vs local equilibrium (section 3), the correlation between diffusion coordinates and the commitor function (section 4). Finally, the authors illustrate those properties on several standard toy models including a system with double-well potential, alanine dipeptide and deca-alanine (section 5). Although the manuscript is well-written and has some potential, I don't think that the presented results provide any relevant contribution to the field, as I am explaining in the attached file. Furthermore, I think some of the statements made about diffusion maps, e.g. the ability of diffusion maps to account for dynamical properties of molecular systems, deserve better discussion and investigation (see file attached) as those involve tricky questions already investigated by several authors. In my opinion, investigating those crucial aspects of diffusion maps would provide a good contribution to the field. However, this would require too major revision to the current manuscript which would then result in a completely different paper. For those reasons, I think the current manuscript is not suitable for publication in RSPA.

Decision letter (RSPA-2019-0036.R0)

08-May-2019

Dear Dr Trstanova

The Editor of Proceedings A has now received comments from referees on the above paper and would like you to revise it in accordance with their suggestions which can be found below (not including confidential reports to the Editor).

Please submit a copy of your revised paper within four weeks - if we do not hear from you within this time then it will be assumed that the paper has been withdrawn. In exceptional circumstances, extensions may be possible if agreed with the Editorial Office in advance.

Please note that it is the editorial policy of Proceedings A to offer authors one round of revision in which to address changes requested by referees. If the revisions are not considered satisfactory by the Editor, then the paper will be rejected, and not considered further for publication by the journal. In the event that the author chooses not to address a referee's comments, and no scientific justification is included in their cover letter for this omission, it is at the discretion of the Editor whether to continue considering the manuscript.

- Acknowledgements
- Funding statement

To revise your manuscript, log into <http://mc.manuscriptcentral.com/prsa> and enter your Author Centre, where you will find your manuscript title listed under "Manuscripts with Decisions." Under "Actions," click on "Create a Revision." Your manuscript number has been appended to denote a revision.

You will be unable to make your revisions on the originally submitted version of the manuscript. Instead, revise your manuscript and upload a new version through your Author Centre.

When submitting your revised manuscript, you will be able to respond to the comments made by the referee(s) and upload a file "Response to Referees" in "Section 6 - File Upload". Please use this to document how you have responded to the comments, and the adjustments you have made. In order to expedite the processing of the revised manuscript, please be as specific as possible in your response to the referee(s).

IMPORTANT: Your original files are available to you when you upload your revised manuscript. Please delete any unnecessary previous files before uploading your revised version.

When revising your paper please ensure that it remains under 28 pages long. In addition, any pages over 20 will be subject to a charge (£150 + VAT (where applicable) per page). Your paper has been ESTIMATED to be 20 pages.

Once again, thank you for submitting your manuscript to Proc. R. Soc. A and I look forward to receiving your revision. If you have any questions at all, please do not hesitate to get in touch.

Yours sincerely
Raminder Shergill
proceedingsa@royalsociety.org

on behalf of
Dr Anders Hansen
Board Member
Proceedings A

Reviewer(s)' Comments to Author:

Referee: 1

Comments to the Author(s)
See review attached

Referee: 2

Comments to the Author(s)
Authors of the manuscript with ID RSPA-2019-0036 have proposed an approach based on diffusion maps for identifying the metastable sets and the corresponding committor functions (in line with previous similar attempts from the literature such as the herein refs 9 and 40). The approach is well discussed and convincing examples are presented (both toy models and more complex cases). This Referee thinks that the works makes a very good contribution to the field and deserves publication as is.

Referee: 3

Comments to the Author(s)
See attached file

Referee: 4

Comments to the Author(s)

The manuscript by Trstanova and coworkers is about Diffusion Map, a dimensionality reduction technique first introduced by Coifman and Lafon in 2006, intended to identify meaningful, i.e., slowly-varying collective variables of large complex systems. Diffusion map was originally shown to approximate the spatial eigenvectors of the Fokker-Planck equation that describes the dynamics of brownian systems such as proteins and other molecular systems in a water environment. As mentioned by the authors in the introduction, an important application of Diffusion Map is the construction of adaptive sampling methods which are based on the biasing of molecular dynamics towards meaningful coordinates – in the present case, the diffusion coordinates – which enable the exploration of the conformational space in a more efficient way. Such methods are, in my opinion, crucial as standard molecular dynamics performed on regular CPUs are unable to explore relevant transitions or conformational changes of realistic protein systems in a reasonable amount of time.

In the current manuscript, Trstanova and coworkers explore different aspects of diffusion maps (and TM diffusion maps) including mathematical aspects (section 2), properties of diffusion maps when constructed upon global vs local equilibrium (section 3), the correlation between diffusion coordinates and the commitor function (section 4). Finally, the authors illustrate those properties on several standard toy models including a system with double-well potential, alanine dipeptide and deca-alanine (section 5). Although the manuscript is well-written and has some potential, I don't think that the presented results provide any relevant contribution to the field, as I am explaining in the attached file. Furthermore, I think some of the statements made about diffusion maps, e.g, the ability of diffusion maps to account for dynamical properties of molecular systems, deserve better discussion and investigation (see file attached) as those involve tricky questions already investigated by several authors. In my opinion, investigating those crucial aspects of diffusion maps would provide a good contribution to the field. However, this would require too major revision to the current manuscript which would then result in a completely different paper. For those reasons, I think the current manuscript is not suitable for publication in RSPA.

Board member pre-assessment comments (if available):

Board Member: 1

Comments to Author(s):

(There are no comments.)

Board Member: 2

Comments to Author(s):

(There are no comments.)

Author's Response to Decision Letter for (RSPA-2019-0036.R0)

See Appendix A.

RSPA-2019-0036.R1 (Revision)

Review form: Referee 1

Quality of the paper

Good

Can the paper be shortened without overall detriment to the main message?

Yes

Do you think some of the material would be more appropriate as an electronic appendix?

No

For papers with colour figures - is colour essential?

Yes

If there is supplementary material, is this adequate and clear?

Not applicable

Are there details of how to obtain materials and data, including any restrictions that may apply?

Yes

Do you have any ethical concerns with this paper?

No

Recommendation?

Accept as is

Comments to the Author(s)

There is one broken reference on page 12, line 54

Review form: Referee 3

Quality of the paper

Good

Can the paper be shortened without overall detriment to the main message?

Yes

Do you think some of the material would be more appropriate as an electronic appendix?

No

For papers with colour figures - is colour essential?

Yes

If there is supplementary material, is this adequate and clear?

Not applicable

Are there details of how to obtain materials and data, including any restrictions that may apply?

Yes

Do you have any ethical concerns with this paper?

No

Recommendation?

Major revision is needed (please make suggestions in comments)

Comments to the Author(s)

The authors addressed my concerns but I have some problem with the new section 5b where a basic adaptive sampling algorithm is presented. In particular:

- on p.18, last paragraph, the authors say that after exploring the QSD within the left metastable state of Ala2, they identify two dihedral angles as CVs to guide the adaptive sampling. But these two dihedrals involve only hydrogen and nitrogen atoms, and should be unrelated to any of the slow dynamics. I wonder why they do not find the psi-angle as representative of the slow dynamics within the left metastable state?

- I doubt that applying metadynamics along these collective variables will speed up the sampling. In fact, I also wonder if metadynamics along the psi-angle would help, as it is kind of orthogonal to the slowest process.

- I don't understand the metric they use in Fig. 16 on p.19. In the text, it says they measure the "RMSD difference between the dihedral angle values of the initial state and the sampled states." I don't know what that means. Why do they not simply track the phi-angle as a function of time?

These points need to be addressed. In addition, I'm a bit surprised by Fig. 9A, where samples of Ala2 are colored by the value of the first diffusion coordinate. But there are only a few spurious negative values, the rest is positive. This does not look like a good representation of the slowest process. I did not notice this when first reading the manuscript.

Review form: Referee 4

Quality of the paper

Good

Can the paper be shortened without overall detriment to the main message?

Yes

Do you think some of the material would be more appropriate as an electronic appendix?

No

For papers with colour figures - is colour essential?

Yes

If there is supplementary material, is this adequate and clear?

Not applicable

Are there details of how to obtain materials and data, including any restrictions that may apply?

Yes

Do you have any ethical concerns with this paper?

No

Recommendation?

Accept as is

Comments to the Author(s)

The authors took my comments into account and made modifications to the manuscript accordingly. I'm especially happy about the new section where an enhanced sampling algorithm (with a proof-of-concept) is presented which shows the practicality and applicability of their mathematical investigation and give ideas for future development in the field.

Decision letter (RSPA-2019-0036.R1)

18-Sep-2019

Dear Dr Trstanova

The Editor of Proceedings A has now received comments from referees on the above paper and would like you to revise it in accordance with their suggestions which can be found below (not including confidential reports to the Editor).

Please submit a copy of your revised paper within four weeks - if we do not hear from you within this time then it will be assumed that the paper has been withdrawn. In exceptional circumstances, extensions may be possible if agreed with the Editorial Office in advance.

Please note that it is the editorial policy of Proceedings A to offer authors one round of revision in which to address changes requested by referees. If the revisions are not considered satisfactory by the Editor, then the paper will be rejected, and not considered further for publication by the journal. In the event that the author chooses not to address a referee's comments, and no scientific justification is included in their cover letter for this omission, it is at the discretion of the Editor whether to continue considering the manuscript.

- Acknowledgements
- Funding statement

To revise your manuscript, log into <http://mc.manuscriptcentral.com/prsa> and enter your Author Centre, where you will find your manuscript title listed under "Manuscripts with Decisions." Under "Actions," click on "Create a Revision." Your manuscript number has been appended to denote a revision.

You will be unable to make your revisions on the originally submitted version of the manuscript. Instead, revise your manuscript and upload a new version through your Author Centre.

When submitting your revised manuscript, you will be able to respond to the comments made by the referee(s) and upload a file "Response to Referees" in "Section 6 - File Upload". Please use this to document how you have responded to the comments, and the adjustments you have made. In order to expedite the processing of the revised manuscript, please be as specific as possible in your response to the referee(s).

IMPORTANT: Your original files are available to you when you upload your revised manuscript. Please delete any unnecessary previous files before uploading your revised version.

When revising your paper please ensure that it remains under 28 pages long. In addition, any pages over 20 will be subject to a charge (£150 + VAT (where applicable) per page). Your paper has been ESTIMATED to be 24 pages.

Once again, thank you for submitting your manuscript to Proc. R. Soc. A and I look forward to receiving your revision. If you have any questions at all, please do not hesitate to get in touch.

Yours sincerely
Raminder Shergill
proceedingsa@royalsociety.org

on behalf of
Dr Anders Hansen
Board Member
Proceedings A

Reviewer(s)' Comments to Author:
Referee: 1

Comments to the Author(s)
There is one broken reference on page 12, line 54

Referee: 3

Comments to the Author(s)
The authors addressed my concerns but I have some problem with the new section 5b where a basic adaptive sampling algorithm is presented. In particular:

- on p.18, last paragraph, the authors say that after exploring the QSD within the left metastable state of Ala2, they identify two dihedral angles as CVs to guide the adaptive sampling. But these two dihedrals involve only hydrogen and nitrogen atoms, and should be unrelated to any of the slow dynamics. I wonder why they do not find the psi-angle as representative of the slow dynamics within the left metastable state?

- I doubt that applying metadynamics along these collective variables will speed up the sampling. In fact, I also wonder if metadynamics along the psi-angle would help, as it is kind of orthogonal to the slowest process.

- I don't understand the metric they use in Fig. 16 on p.19. In the text, it says they measure the "RMSD difference between the dihedral angle values of the initial state and the sampled states." I don't know what that means. Why do they not simply track the phi-angle as a function of time?

These points need to be addressed. In addition, I'm a bit surprised by Fig. 9A, where samples of Ala2 are colored by the value of the first diffusion coordinate. But there are only a few spurious negative values, the rest is positive. This does not look like a good representation of the slowest process. I did not notice this when first reading the manuscript.

Referee: 4

Comments to the Author(s)

The authors took my comments into account and made modifications to the manuscript accordingly. I'm especially happy about the new section where an enhanced sampling algorithm (with a proof-of-concept) is presented which shows the practicality and applicability of their mathematical investigation and give ideas for future development in the field.

Board member pre-assessment comments (if available):

Board Member

Comments to Author(s):

(There are no comments.)

Author's Response to Decision Letter for (RSPA-2019-0036.R1)

See Appendix B.

RSPA-2019-0036.R2 (Revision)

Review form: Referee 3

Quality of the paper

Good

Can the paper be shortened without overall detriment to the main message?

Yes

Do you think some of the material would be more appropriate as an electronic appendix?

No

For papers with colour figures – is colour essential?

Yes

If there is supplementary material, is this adequate and clear?

Not applicable

Are there details of how to obtain materials and data, including any restrictions that may apply?

Yes

Do you have any ethical concerns with this paper?

No

Recommendation?

Accept as is

Comments to the Author(s)

The authors have addressed all my concern and I recommend the manuscript for publication.

Decision letter (RSPA-2019-0036.R2)

Dear Dr Trstanova

I am pleased to inform you that your manuscript entitled "Local and Global Perspectives on Diffusion Maps in the Analysis of Molecular Systems" has been accepted in its final form for publication in Proceedings A.

Our Production Office will be in contact with you in due course. You can expect to receive a proof of your article soon. Please contact the office to let us know if you are likely to be away from e-mail in the near future. If you do not notify us and comments are not received within 5 days of sending the proof, we may publish the paper as it stands.

Open access

You are invited to opt for open access, our author pays publishing model. Payment of open access fees will enable your article to be made freely available via the Royal Society website as soon as it is ready for publication. For more information about open access please visit http://royalsocietypublishing.org/site/authors/open_access.xhtml. The open access fee for this journal is £1700/\$2380/€2040 per article. VAT will be charged where applicable.

Note that if you have opted for open access then payment will be required before the article is published – payment instructions will follow shortly.

If you wish to opt for open access then please inform the editorial office (proceedingsa@royalsociety.org) as soon as possible.

Your article has been estimated as being 23 pages long. Our Production Office will inform you of the exact length at the proof stage.

Proceedings A levies charges for articles which exceed 20 printed pages. (based upon approximately 540 words or 2 figures per page). Articles exceeding this limit will incur page charges of £150 per page or part page, plus VAT (where applicable).

Under the terms of our licence to publish you may post the author generated postprint (ie. your accepted version not the final typeset version) of your manuscript at any time and this can be made freely available. Postprints can be deposited on a personal or institutional website, or a

recognised server/repository. Please note however, that the reporting of postprints is subject to a media embargo, and that the status the manuscript should be made clear. Upon publication of the definitive version on the publisher's site, full details and a link should be added.

You can cite the article in advance of publication using its DOI. The DOI will take the form: 10.1098/rspa.XXXX.YYYY, where XXXX and YYYY are the last 8 digits of your manuscript number (eg. if your manuscript number is RSPA-2017-1234 the DOI would be 10.1098/rspa.2017.1234).

For tips on promoting your accepted paper see our blog post:
<https://blogs.royalsociety.org/publishing/promoting-your-latest-paper-and-tracking-your-results/>

On behalf of the Editor of Proceedings A, we look forward to your continued contributions to the Journal.

Sincerely,

Raminder Shergill
proceedingsa@royalsociety.org

Appendix A

Zofia Trstanova

✉ zofia.trstanova@gmail.com

Proceedings of the Royal Society

Response to Referee's Comments, RSPA-2019-0036

(Local and Global Perspectives on Diffusion Maps

in the Analysis of Molecular Systems by Z.

Trstanova, B. Leimkuhler, and T. Lelievre.)

August 7, 2019

Dear Editors,

We thank the reviewers for their careful reading of the manuscript and their comments. We address their detailed concerns in the revised draft and, sequentially, in the commentary provided below.

We emphasize that the major modification of the article is the additional section, "From local to global: defining metastable states and enhanced sampling".

Referee #1

General Comment

My main recommendation is to re-structure the manuscript... We have re-structured the manuscript as requested by the referee by gathering the results on the committor in Section 4, and introducing a new section on enhanced sampling.

Detailed Comments

- Can the authors provide more details on how to obtain the formula in Remark 2.3?*
This formula is the Nyström extension [1], we have extended the text in the references.
- Section 3, "Quasi-stationary distribution": why is the domain a subset of RdN ? What is N ?*
We thank the referee for pointing out this typo, the space is indeed \mathbb{R}^d .
- Same paragraph: is it also possible to extract the QSD itself using diffusion maps and the modifications for boundary value problems? Can the authors show numerical estimates of the QSD obtained from diffusion maps in some of the examples?*
Yes, this is indeed possible. We have performed these experiments, but in the interest of brevity we opted for not presenting it.
- Same paragraph, sentence starting "we compute the statistical averages of various observables ...": please provide more details on what you are doing here, and maybe even include some results in the figure. This is an important step.*
This step is important to detect convergence to local equilibrium. The accuracy of diffusion maps depends also on the statistical error (Central Limit Theorem), so we want to generate trajectories which are long enough to resolve the stationary state. We did not include such figures for brevity at this stage. Please see the added section 5, where we precisely describe the strategy for detecting local equilibrium.
- Figure 6: this figure deserves more explanation. What happens to EV_0 after the second minimum is explored by the simulation? Should it not be equal to zero from this point on? Also, it seems the transition of EV_1 to the rate of transition over the main barrier is almost instantaneous, whereas the faster rates take much longer to adjust. That is confusing. I*

would suggest to focus on only the first two or three eigenvalues and choose a better scaling to illustrate what is happening in detail.

This figure has been suppressed and Section 5 gives more aspects and details on this.

6. *Section 3, "Algorithmic identification of metastable subsets": I think PCCA has become the standard method to perform clustering based on the eigenvectors. Is there a reason the authors did not use PCCA?*

The referee is right in suggesting the use of PCCA, however we found that it is possible to obtain the metastable states by clustering the dominant eigenvectors of the diffusion maps, which is a much simpler approach as we have a direct approximation of the generator of the dynamics and therefore do not need to choose "discretization boxes" as in [2].

7. *Alanine dipeptide: why did the authors use TMDmap if the data set is already equilibrated?*

TMDmap is an extension of the vanilla diffusion map based on weighted ensembles of samples and importance sampling. We agree that TMDmap matters more in later calculations of the article, in particular in Sections 4b and 5.

8. *Deca alanine: I do not believe it takes 100 microseconds to equilibrate simulations of deca alanine. Most analyses I have seen were using data sets of five to ten microseconds total simulation time.*

We thank the referee for pointing out this reference mistake, we have corrected the simulation time. Please note that we have not run the full simulation for this paper, but used an enhanced sampling trajectory obtained by Infinite Simulated Swap Tempering (ISST).

9. *Deca alanine: can the authors also show the results for the application of plain diffusion maps to the biased simulation data, to illustrate the improvement due to the use of TMDmap?*

The analyzed trajectory consisted of samples from an ISST sampling and weights for a target temperature. In order to unbiased the sampling w.r.t. to the Boltzmann distribution at a given temperature, one needs to use the weights, which is only possible with the TMDmap, but not with the vanilla diffusion map. Showing results with vanilla map would not really make sense because it would not be able to unbiased the sampling.

Referee #2

No response needed.

Referee #3

General Comments

First of all, I would recommend that the Authors do a more thorough job at carefully indicating their own contribution...

We have done this in a number of places and hope this is now more clear. We have also attempted to improve the presentation of relationship of the current work to the several works mentioned by the referee.

..and in particular how they improved or modified literature results that are stated, such as Refs [24] (committor calculation) and 26 (mathematical characterization of quasistationary distribution).

With regard (the former reference) [24] (Lai and Lu 2018) we clarified the relation to our work in the introduction, namely pointing out that this paper only considered toy models and there are many issues that enter the discussion of more complicated systems, as we have discussed in the paper. In that paper, there is also no indication of how the localized diffusion maps would relate to sampling of extended metastable systems. The papers of Clementi et al (refs 40,52 of the original submissions) are closer in spirit to the current work, and helped to guide us, but the procedures describe there are more heuristic. We have discussed the use of the QSD as a tool for example to get a converged local sampling, something that was not considered in Clementi's papers.

Second, the Authors should address the problem of choosing an appropriate localscale in the Diffusion Map kernels; that is a parametric choice which is known to seriously influence the performance of the results.

We looked carefully for evidence of the relevance of the local scaling in the diffusion map kernel, as noted in the works of Clementi, but were unable to show it in our experiments. This could be due to many factors, e.g. differences in the particular systems studied; we therefore prefer not to address this issue in the current work.

Detailed Comments

1. *page 3, from line 11 on. Miss some references on recent Diffusion Map developments, such as Ref. [40] (where local scale is computed using k nearest neighbors) and Boninsegna et al. J. Chem. Theory Comput. (2015), where Diffusion Map results are optimized using a variational formulation.*

We have added the missing reference.

2. *page 4, line 14. "We make a connection to the diffusion map with IS formalized" (singular).* We thank the reviewer, we have corrected this.

3. *page 4, lines 15-16. Limitations is used twice in the same sentence.*

We thank the reviewer, we have corrected this.

4. *page 4, Langevin dynamics and Boltzmann distribution. Mention that $V(x)$ is the potential energy driving the diffusion process.*

We have extended the text.

5. *page 5, lines 28-30. It would be beneficial to the novice reader, if the different steps of the algorithm in going from h_ε to P_ε were written out explicitly; for instance, the matrix $P_{\varepsilon,\alpha}$ is only mentioned.*

We apologize that we have forgotten to omit the α in $P_{\varepsilon,\alpha}$, which might have confused the reader. All the necessary steps are listed after "To be precise, ..." and also the explicit definition of $P_{\varepsilon,\alpha}$ follows.

6. *page 6, lines 10 to 12. To the best of my knowledge, extracting physical timescales from Diffusion Map eigenvalues is a non-trivial issue. Could the Authors comment on that? Also, the inequality chain $0 = \lambda_0 > \lambda_1 \geq \dots$ is inconsistent with that on page 4, line 54 ($\lambda_0 = 0 > -\lambda_1 \geq \dots$), notation-wise.*

We have adjusted the notation. We are aware that the generator of overdamped Langevin dynamics cannot provide physical timescales of underdamped Langevin dynamics. We have addressed this issue in Remark 2.2. However, in order to prevent the confusion of the reader, we remove the mentioned sentence.

With regard to extraction of physical timescale information, we point the referee to our comments in the conclusion which provide a possible way forward in computations of this

nature (although the details are beyond the scope of the current article).

7. *page 6, eq (2.12) It would be helpful to explicitly mention that p_t are the conjugate momenta, and define the entries of the mass matrix.*

We have addressed this.

8. *page 6, line 40. Could the Authors be more specific as to what consistent discretization means? An example or a reference would be helpful.*

We had in mind ergodic numerical schemes as for example BAOAB integrator (Leimkuhler, Matthews, 2013).

9. *page 6, from line 55 on. it would be beneficial to the novice reader if the different steps were written out explicitly, for instance by using a bullet point style (but this is my personal taste).*

We prefer not to do this, in the interests of compactness. This part of the article is directly analogous to more detailed treatments in other previous articles and is merely presented for completeness.

10. *page 7, Dirichlet boundary problems. Some procedure details are missing, such as which is the x range explored by the simulation points? Which of those values constitute the "points outside Ω "?*

We have extended this section to add more details.

11. *page 8, eq (3.1) Is it \mathcal{L} or \mathcal{L}^* ? It looks it should be the latter, according to ref. [26], eq (8).*

This is actually correct as written. Notice that in (3.2), the QSD is proportional to the product of the first eigenvector of \mathcal{L} with $\exp(-\beta V)$. It can be checked that this is equivalent to the fact that the QSD is proportional to the first eigenvector of \mathcal{L}^* (again with Dirichlet boundary conditions).

12. *page 9, Fig. 3. Use one legend with a consistent font.*

We have adjusted the figure as suggested.

13. *page 9, line 43. Could the Authors provide the mathematical expression for both configurational and kinetic temperatures?*

We have expanded the text as requested.

14. *page 10, section 4. Add references to more methods to identify metastable states, such as PCCA (and variants), and comment on which information identifying metastable states provides.*

We have extended the section with more references.

15. *page 10, line 55. Notation inconsistency: here Ω indicates the full configurational space, but in the previous section indicated metastable sets.*

We have adjusted the notation following the suggestions of the referee.

16. *page 11, lines 57 on. It is not fully clear where and why the choice of the number of nearest neighbors plays a role in the procedure. Also, which pseudo-spectral method was used to solve the Dirichlet problem?*

The number of nearest neighbors is a parameter in the method which can impact rather the computational complexity than the accuracy of the diffusion map.

Could the Authors provide more details on how the coordinate discretization was performed?

In the diffusion map case, the "coordinate discretization" was performed by generating Markov chain samples. In the case of the pseudo-spectral method, a grid was chosen very fine to keep the evaluation error small enough.

It would be interesting to look at the method robustness as the definition of the metastable states change, e.g., by making them larger and larger.

Yes, indeed, we have performed the analysis, but decided not include the results as their

analysis would be involved and ultimately detract from the presentation of the main idea of the paper. The results suggest that the method is robust with respect to the variation of the definition domains, under the assumption that the domains A and B are within the metastable state.

17. *page 12, line 28. I believe the transfer operator P_τ is mentioned for the first time, and its eigenvalues are also labeled with λ as the L eigenvalues. To avoid confusion, I would discuss that on page 4, in the Langevin dynamics and Boltzmann distribution section, and point out that the two sets of eigenvalues are related by an exponentiation. I also feel that the whole paragraph would benefit from rewriting to make it more quantitative: there are a few vague notions such as fixed threshold or eigenvalue close to 1. The algorithm steps need to be explained more rigorously, to make it transferrable.*

We have provided substantially more details on this topic in Section 5.

18. *page 13, line 53. Could the Authors briefly comment on how they used nonlinear regression? Alternatively, if the procedure they use is well established, could they provide a reference?*

We have extended the text with more information.

19. *page 14, line 47. I believe that Ala12 (not Ala10) is discussed in ref [40].*

We thank the referee for pointing out this mistake.

20. *page 16, Fig. 14. I recommend that configuration cartoons from Fig. 1 be added on the top of the free energy plots, for the sake of clarity. Also, the intrinsic change in the slowest process as the temperature increases seems very interesting. Could the Authors comment more on that, perhaps by identifying the intermediates/transition states and characteristic timescales associated with that? Are the results consistent with other literature findings?*

Identifying the timescales and transition states for the underdamped Langevin dynamics would require substantially more work. The diffusion map analysis gives good CVs, metastable states and an approximation of the committor, which could then be used in other numerical methods to perform such analysis (for example Forward Flux Sampling or Adaptive Multilevel Splitting).

21. *page 16, Data Accessibility. Add references to software used for producing the results. Are the results accessible free of charge?*

We have published the datasets. The software used is OpenMM and pydiffmap.

This information is provided at the end of the manuscript.

22. *page 16, Authors contributions missing.*

We have provided this information to the editor, it will be included in the final version.

23. *page 19, journal title is missing from ref [27].*

We have updated the reference.

Referee #4

General Comments

We appreciate this thorough review which we attempt to address constructively below.

...Although the mathematical investigation of diffusion maps upon quasi-equilibrium is new (p.8), this analysis is, in my opinion, not useful as current sampling methods based on local diffusion maps seem to work just fine.

We must take exception with the philosophy espoused here, namely that mathematical treatment is not useful if methods "seem to work just fine." The mathematical treatment can

explain scientific practice, to justify it, or to raise new questions about its validity or potential expansion. There is no sense in which diffusion maps are ‘done and dusted’—there is in fact no currently established sampling framework based on these ideas built in to the current leading edge molecular software systems, this because many of the techniques still require refinement. This paper helps with these foundations.

To give a specific example, our analysis explains why the previous methods worked in the setting of a not fully explored distribution: the quasi-stationary distribution explains the convergence of diffusion map approximations within the metastable state.

...for example, the authors could have provided some explanation/discussion on how to improve existing sampling methods based on their mathematical investigation.

We have extended the discussions of relevance and positioning of our work throughout the paper (as discussed in responses to the other referees). Moreover, we have now added a whole section which describes our envisioned algorithm for enhanced sampling, along with a proof-of-concept simulation. More detailed evaluation will need to await further research (which the authors are currently engaged in performing).

Regarding point 2: although mathematically elegant, the authors did not explain the importance and practical aspects of investigating the relationship between the first eigenvector and the commitor. Again, can we use that information to design better adaptive sampling methods?

This is now addressed in the new section, where we explore the relevance of the learned eigenvectors for enhanced sampling procedures. The idea is that one can identify a state by looking at the spectrum computed by diffusion map. Once states are thus implicitly defined, one can use enhanced sampling techniques (adaptive biasing or accelerated dynamics technique a la AF Voter) to exit from these metastable states. This is illustrated in Section 5 using metadynamics.

We admit that further investigation will be needed to fully assess the procedures, but we do offer a precisely stated algorithm which can frame such studies.

...can we practically compute the reaction rates from the diffusion coordinates or diffusion map eigenvalues? This is, in my opinion, a central aspect that should be further investigated...

Diffusion maps provide approximation of overdamped Langevin dynamics, which is close but still different from finite friction underdamped Langevin dynamics, i.e. the usual molecular dynamics sampler for complex molecules. Dynamical properties such as transition rates are therefore different.

Although this question falls outside the scope of our article, which is aimed at mathematical foundations of enhanced sampling procedures, we discuss, in the conclusion, a possible way to address dynamical quantities such as exit times. Further work is definitely needed to assess the potential of such methods.

Detailed Comments

1. *p.2 line 14: The authors mentioned that the “The difficulty arises due to the high-dimensional, multimodal nature of the target distribution, in which the high-likelihood conformational states are separated by low-probability transition regions. In such a setting, the transitions between critical states become “rare events”, meaning that naive*

computational approaches converge slowly (if at all) or produce inaccurate results.” Although those statements are true, I would also mention the fact that standard MD simulations of large proteins (>500 residues) run on regular CPUs allow to simulate dynamics over hundreds of ns (or a few μs max) in a reasonable amount of time. This time scale is often not enough to sample relevant conformational transitions which may occur on the ms time scale. So, in the end, the difficulty is not only because of low-probability transition regions but also because of limited computational resources.

We have modified the paragraph as follows: "The difficulty arises due to the high-dimensional, multimodal nature of the target distribution in combination with limited computational resources. The multimodality of the target distribution causes that the high-likelihood conformational states are separated by low-probability transition regions. In such a setting, the transitions between critical states become “rare events”, meaning that naive computational approaches converge slowly (if at all) or produce inaccurate results. Moreover, standard MD simulations of large proteins (>500 residues) run on regular CPUs allow to simulate dynamics over hundreds of ns (or a few μs max) in a reasonable amount of time. This time scale is often not enough to sample relevant conformational transitions which may occur on the ms time scale. "

2. p. 2, line 23: “the results of Langevin dynamics simulations” should be changed by “the results of short Langevin dynamics simulations”

We have modified the text.

3. p2. line 49: “The methods discussed here have”. I’m not sure what methods the authors are referring to.

We refer to the qsd diffusion map, and the final section.

4. p.5 line 8-12: it seems to me that the diffusion map coordinates also include temporality via the exponential factors appearing in Eq. (2.5). So does the diffusion distance. See for example Eq. (3.3) in “diffusion maps, reduction coordinates and low dimensional representation of stochastic systems” from Coifman. Please explain.

The exponential factors indeed include temporality for the overdamped Langevin dynamics.

5. p.5, line 22-23: it is mentioned that the norm in Eq. 2.6 should correspond to the Euclidean norm in the configurations space. But this is in contradiction with the work of many other authors that tried many different metrics including RMSD (see: Boninsegna, L., Gobbo, G., Noé, F., Clementi, C. (2015). Investigating molecular kinetics by variationally optimized diffusion maps. *Journal of chemical theory and computation*, 11(12), 5947-5960.). This should be made clear by the authors.

The referee is right, we have formulated the statement in more general setting.

6. p.7 line 33: One or two references should be given to accompany the equation in remark (2.3) as this equation is the Nyström extension and has been widely used in many contexts. See for example: [Y. Bengio et al 2004].

We have added the reference.

Publications

Yoshua Bengio, Olivier Delalleau, Nicolas Le Roux, Jean-François Paiement, Pascal Vincent, and Marie Ouimet. Learning eigenfunctions links spectral embedding and kernel PCA. *Neural computation*, 16(10):2197–2219, 2004.

Ch Schütte, Alexander Fischer, Wilhelm Huisinga, and Peter Deuffhard. A

direct approach to conformational dynamics based on hybrid monte carlo.
Journal of Computational Physics, 151(1):146–168, 1999.

Appendix B

Zofia Trstanova

✉ zofia.trstanova@gmail.com

Proceedings of the Royal Society

Response to Referee's Comments, RSPA-2019-0036

(Local and Global Perspectives on Diffusion Maps

in the Analysis of Molecular Systems by Z.

Trstanova, B. Leimkuhler, and T. Lelievre.)

October 17, 2019

Dear Editors,

We thank the reviewers for their careful revision of the manuscript and their comments. We address the additional comments of referee #3 below.

Response to referee #1

There is one broken reference on page 12, line 54

We have fixed the reference.

Response to referee #3

The authors addressed my concerns but I have some problem with the new section 5b where a basic adaptive sampling algorithm is presented. In particular:

- 1. On p.18, last paragraph, the authors say that after exploring the QSD within the left metastable state of Ala2, they identify two dihedral angles as CVs to guide the adaptive sampling. But these two dihedrals involve only hydrogen and nitrogen atoms, and should be unrelated to any of the slow dynamics. I wonder why they do not find the psi-angle as representative of the slow dynamics within the left metastable state? I doubt that applying metadynamics along these collective variables will speed up the sampling. In fact, I also wonder if metadynamics along the psi-angle would help, as it is kind of orthogonal to the slowest process.*
We have indeed observed that the ϕ, ψ angles have high correlations in agreement with existing literature (we have however found also other angles which have high correlations with diffusion maps obtained within the first metastable state). In order to improve interpretability of our results, we have simplified the list of atoms on which we compute the dihedral angles by removing the hydrogen atoms from the possible quadruples. We find that ϕ, ψ are correlated, however the highest correlation can be found in the combination of ACE1C-ALA2C-ALA2C-NME3N and ACE1C-ACE1O-ALA2N-NME3N. We modified the Figure 16 to compare metadynamics with these angles and ϕ, ψ . We stress, that we do not expect that the slowest modes of the first metastable state are the same as the global slowest modes. Finally, the physical collective variables which correlate with the diffusion map coordinates are not unique.
- 2. I don't understand the metric they use in Fig. 16 on p.19. In the text, it says they measure the "RMSD difference between the dihedral angle values of the initial state and the sampled states." I don't know what that means. Why do they not simply track the phi-angle as a function of time?*

We have replaced this exploration metric with tracking the phi-angle over time which is more common in the literature.

3. In addition, I'm a bit surprised by Fig. 9A, where samples of Ala2 are colored by the value of the first diffusion coordinate. But there are only a few spurious negative values, the rest is positive. This does not look like a good representation of the slowest process. I did not notice this when first reading the manuscript.

This might be due to the colorbar, we have reverified our results and diffusion maps indeed separate the two metastable states with diffusion map coordinates correlating with ϕ, ψ angles.

Long trajectory of alanine dipeptide: diffusion map coordinates parameterize the slowest process. We have emphasized the positive values of DC1 with black circles around the points and negative values with blue circles. We only plot 2000 trajectory points.